# Manifold-Aware Perturbations for Constrained Generative Modeling

**Katherine Keegan** [1]   **Lars Ruthotto** [1]

## Abstract

Generative models have enjoyed widespread success in a variety of applications. However, they encounter inherent mathematical limitations in modeling distributions where samples are constrained by equalities, as is frequently the setting in scientific domains. In this work, we develop a computationally cheap, mathematically justified, and highly flexible distributional modification for combating known pitfalls in equality-constrained generative models. We propose perturbing the data distribution in a constraint-aware way such that the new distribution has support matching the ambient space dimension while still implicitly incorporating underlying manifold geometry. Through theoretical analyses and empirical evidence on several representative tasks, we illustrate that our approach consistently enables data distribution recovery and stable sampling with both diffusion models and normalizing flows.

## 1. Introduction

Generative models, such as Denoising Diffusion Probabilistic Models (DDPMs) (Ho et al., 2020), Score-Based Diffusion Models (SBDMs) (Song et al., 2021), and Normalizing Flows (NFs) (Papamakarios et al., 2021), have become ubiquitous for their ability to obtain high-quality samples from complex unknown distributions given finite data. These state-of-the-art paradigms model the data distribution $p_0$ as a pushforward of a tractable latent distribution $p_T$ (often Gaussian) under some function $f$, which are then connected via the change of variables formula

$$p_T(x) = p_0(f^{-1}(x))|\det J_f(x)|^{-1}. \qquad (1)$$

The terminal Gaussian requirement ensures that the support of $p_T$ in $\mathbb{R}^d$ is *non-degenerate*, or fully $d$-dimensional. However, in many scientific tasks, samples $x \sim p_0$ also satisfy

some $h(x) = 0$, $h : \mathbb{R}^d \to \mathbb{R}^{d-m}$, often arising from physical laws or domain knowledge. The presence of an equality constraint with full-rank Jacobians means that $p_0$ lies on a $m$-dimensional manifold $\mathcal{M}$, making it a degenerate distribution embedded in $\mathbb{R}^d$. To be precise, $p_0$ is singular with respect to the $d$-dimensional Lebesgue measure.

Generative models under this framework require that the supports of $p_0$ and $p_T$ have equal dimension. NFs rely on invertible transformations between $p_0$ and $p_T$ in Equation (1). With a degenerate $p_0$, the Jacobian determinant will be zero, leading to *exploding log-determinants*. Similarly, in diffusion models on manifold-supported $p_0$, we encounter the issue of *score explosion*, where the score field guiding samples backwards in virtual time $t \in [0, T]$ explodes near the manifold as $t \to 0$, leading to sampling instability, unusable samples, and inaccurate generated distributions.

In this work, we propose a computationally cheap and mathematically motivated modification to $p_0$, making it more amenable to training and sampling under any generative modeling paradigm. Our approach relies on (1) strategically noising data samples into the local normal bundle of $\mathcal{M}$ to form a perturbed distribution $p_\sigma$, where $\sigma$ parametrizes the perturbation strength, (2) training any unconstrained generative model, and then (3) projecting generated samples back to $\mathcal{M}$. We illustrate the utility of this approach on multiple datasets on both simple and complex manifolds. This work shows that our approach offers the following:

- **Perfect distribution recovery** for linear constraints

- **Bounded total variation** for nonlinear constraints

- Guaranteed **constraint adherence**

- **Bounded scores and non-zero Jacobian determinants** for diffusion and NF approaches, respectively, combating known pitfalls in equality-constrained generative model training and sampling arising from numerical instability

- **Improved or competitive** distributional fidelity

This paper shows that for equality-constrained distributions, $p_\sigma$ is an attractive alternative target distribution: it removes the need for constraint adherence during sampling, avoids

---

[1]Department of Mathematics, Emory University, Atlanta, Georgia, USA. Correspondence to: Katherine Keegan <katherine.emiri.keegan@emory.edu>.

*Proceedings of the 43rd International Conference on Machine Learning*, Seoul, South Korea. PMLR 306, 2026. Copyright 2026 by the author(s).

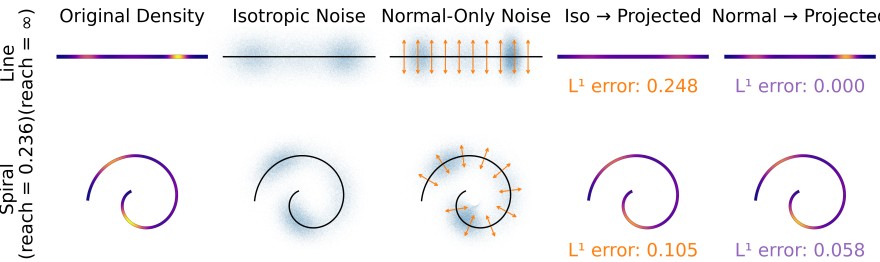

*Figure 1.* An illustration of our proposed manifold-aware perturbation and comparison to isotropic perturbations showing exact recovery for linear manifolds and lower error for the spiral dataset.

score or Jacobian determinant explosion, and offers mathematical guarantees in the induced projected distribution. Our code is available at the following link: `https://github.com/katiekeegan/ManifoldAwarePerturbationsForConstrainedGenerativeModeling`.

## 2. Background and Related Literature

Degeneracy of $p_0(x)$ prevents the existence of a Lebesgue density and leads to several well-known pathologies in generative modeling (Pidstrigach, 2022). For instance, SBDMs assume differentiability of $\log p_0(x)$ in $\mathbb{R}^d$, and NFs require evaluation of log-densities under invertible transforms. Both become ill-defined or unbounded if $p_0(x)$ is singular with respect to the $d$-dimensional Lebesgue measure. In particular, the dimension mismatch of the distributions in Equation (1) prevents the existence of *any* invertible mapping between $p_0(x)$ and $p_T(x)$. In this work, we focus on the case where degeneracy is a result of being constrained by an $m$-dimensional manifold $\mathcal{M}$ with full-rank Jacobians at all points $x \in \mathcal{M}$. The challenge of equality-constrained generative modeling has given rise to a variety of techniques, which may loosely be placed into the following two categories.

**Modifying the underlying distribution.** Many works directly modify the underlying distribution being learned by a generative model, with several techniques employing isotropic noising. This is present in well-known work on improving diffusion model effectiveness when data distributions lie on a lower-dimensional manifold (Song & Ermon, 2019). Similarly, SoftFlow (Kim et al., 2020) adds various levels of Gaussian noise to combat the dimension mismatch between $p_0$ and $p_T$ during NF training and sampling. However, such isotropic approaches *immediately* risk distribution distortion if post-hoc projection is used to ensure constraint adherence, even for the simplest linear constraints.

**Modifying the learning/sampling algorithm.** Other works design novel constrained generative model families, develop constraint-aware training objectives, or enforce constraints during the sampling process. In the first case, many

works focus on the case where the constraints define a Riemannian manifold. In this setting, one can make use of differential geometry and stochastic processes on manifolds to define manifold-intrinsic SDEs, effectively constructing generative models which operate entirely on the constraint surface (Bortoli et al., 2022; Mathieu & Nickel, 2020). While elegantly ensuring constraint satisfaction, deriving such SDEs for general $\mathcal{M}$ is nontrivial. Moreover, Riemannian generative models generally require many nontrivial manifold-aware computations throughout training and sampling, which can be computationally expensive and require careful and efficient implementation (Bortoli et al., 2022). Other recent developments in equality-constrained diffusion modeling include Projected Diffusion Models (PDMs) (Christopher et al., 2024), wherein the learned scores of an unconstrained model are iteratively projected to $\mathcal{M}$ during sampling, and Physics-Informed Diffusion Models (PIDMs) (Bastek et al., 2025), in which constraint residuals are computed at $t = 0$ to form a physics-informed regularization loss term. However, these approaches necessitate high additional computational time: for PDM, this is needed during sampling, and for PIDM, the need to sample during training drastically affects training time, and neither approach offers guarantees of $p_0(x)$ recovery. We remark that our paper bears similarity to an inflation-deflation approach proposed in (Horvat & Pfister, 2023), of which we were unaware during the writing of this paper. However, this previous work focuses on exact density estimation and imposes stricter assumptions accordingly while also restricting their study to the normalizing flow case. Our work may be interpreted as generalizing this fix to constrained generative modeling more broadly, characterizing precise bounds when strict assumptions do not hold, and providing a straightforward off-the-shelf deterministic analogue to their deflation approach with bounded distributional recovery.

## 3. Method

In this paper, we present a novel contribution to works which aim to modify the underlying distribution being learned by a generative model. We propose an elegant distributional modification wherein $p_0$ is strategically distorted in a manifold-

aware manner. In particular, we perturb $p_0(x)$ strictly in the normal direction(s) to $\mathcal{M}$ at each $x$, implicitly incorporating underlying manifold geometry while setting up the modified distribution to have minimal error upon post-hoc manifold projection. Formally, this is as follows:

**Definition 3.1** ($p_\sigma$). Draw $Z \sim p_0$ and $N|Z = z \sim \mathcal{N}(0, \sigma^2 I_{N_z \mathcal{M}})$, where $N_z \mathcal{M}$ denotes the normal space of $\mathcal{M}$ at $z$. Set $X := Z + N \in \mathbb{R}^d$, and write $p_\sigma = \mathrm{Law}(X)$. We refer to $p_\sigma$ as the perturbed or lifted distribution associated with $p_0$ at noise scale $\sigma > 0$.

Operationally, one may implement the formation of $p_\sigma$ by computing local Jacobians of the manifold surface around each sample $x \sim p_0$, using these to obtain basis vectors for the normal space, and perturbing each sample with Gaussian noise in the normal directions. For user reference, the procedure for forming $p_\sigma$ from (numerical or analytic) Jacobians is presented in Algorithm 1.

---

**Algorithm 1** Procedure for forming the modified distribution $p_\sigma$.

---

**Require:** Constraint manifold $\mathcal{M}$, samples $x_0^i \sim p_0$
1: **for** each $x_0^i$ in training data set **do**
2:     Compute Jacobian $J_h(x_0^i)$
3:     Compute orthonormal basis of $J_h(x_0^i)^\top$, e.g. through QR decomposition, to yield orthonormal basis for $N_{x_0^i} \mathcal{M}$
4:     Draw noise $\xi \sim \mathcal{N}(0, \sigma^2 I_k)$
5:     Use basis vectors of $N_{x_0^i} \mathcal{M}$ and map $\xi$ onto the normal space to obtain $\tilde{\xi}$
6:     Construct perturbed sample $x_\sigma^i = x^i + \tilde{\xi}$
7: **end for**

---

The reason for the intentionally anisotropic construction of $p_\sigma$ is twofold: first, noising strictly in normal directions *ensures that $p_\sigma$ is no longer degenerate* in the sense of its support in $\mathbb{R}^d$, vastly improving its compatibility with state-of-the-art (SOTA) out-of-the-box unconstrained generative modeling paradigms. Second, and unique to our work, anisotropic noising in this manner produces limited distortion and reduces artifacts in the induced distribution after nearest-point projection, since tangential distortion along the manifold is minimal in curved regions and zero for linear regions.

To return to $\mathcal{M}$, we consider a (possibly multi-valued) nearest-point projection $\Pi : \mathbb{R}^d \to \mathcal{M}, \Pi(x) \in \arg\min_{z \in \mathcal{M}} \|x - z\|_2$, where ties are broken arbitrarily when the minimizer is not unique.

## 4. Error Analysis

The goal of this section is as follows: first, we discuss the nondegeneracy of $p_\sigma$ and its advantages for generative

---

**Algorithm 2** Our proposed approach for constrained generation by modeling $p_\sigma$.

---

**Require:** Constraint manifold $\mathcal{M}$, samples $x_0^i \sim p_0$
1: Perturb each sample $x_0^i$ strictly in normal direction of $\mathcal{M}$ to obtain $x_\sigma^i \sim p_\sigma$ as in Algorithm 1
2: Train generative model on $\{x_\sigma^i\}$ to model $p_\sigma$
3: Sample $\hat{x}_\sigma^i \sim p_\sigma$
4: Project $\hat{x}_\sigma^i$ to $\mathcal{M}$ using either analytic constraint projector or iterative minimization of $\arg\min_{z \in \mathcal{M}} \|x - z\|_2$

---

modeling. Then, we prove that if constraints are linear, applying $\Pi$ to samples from $p_\sigma(x)$ yields exactly the desired original intrinsic distribution. Finally, we bound the total variation (TV) distance *on the manifold* between the original distribution and the pushforward of $p_\sigma$ upon nearest-point projection. We have constructed this section to be fairly self-contained so that a mathematically-inclined reader may read this section rigorously, while other readers can read the preceding section to understand our approach. For clarity, all variables and abbreviations are also described in the notation glossary provided in Appendix A.

Let $X : \Omega \to \mathbb{R}^d$ define a random variable on a probability space $(\Omega, \mathcal{F}, \mathbb{P})$. Its law is the measure $\mu(A) = \mathbb{P}(X \in A)$, with $A$ being an element of the Borel $\sigma$-algebra $\mathcal{B}(\mathbb{R}^d)$. In order for the law $\mu$ to admit a density, one must select a reference measure $\nu$ on $\mathbb{R}^d$. If $\mu$ is absolutely continuous with respect to $\nu$, then the Radon-Nikodym theorem guarantees the existence of a measurable function $p = \frac{d\mu}{d\nu}$ such that

$$\mu(A) = \int_A p(x) d\nu(x), A \in \mathcal{B}(\mathbb{R}^d).$$

Here, $p$ is interpreted as the *density* of $\mu$ with respect to $\nu$.

The standard reference measure for distributions over Euclidean space is the $d$-dimensional Lebesgue measure $\lambda^d$. A distribution $\mu$ is said to be *full-dimensional* if $\mu \ll \lambda^d$, in which case it admits a Lebesgue density $p(x)$. However, we say a law $\mu$ is *singular* if $\mu$ and $\lambda^d$ are mutually singular, meaning that there exists a measurable set $A \subseteq \mathbb{R}^d$ such that $\mu(A) = 1$ and $\lambda^d(A) = 0$. Singular distributions arise naturally in many applications where data are constrained to lie on a lower-dimensional subset of the ambient space.

In this work, we refer to such singular laws as *degenerate distributions*. A canonical example of degeneracy arises when the support of $\mu$ is contained in an $m$-dimensional subset (e.g. a manifold $\mathcal{M}$) of $\mathbb{R}^d$ with $m < d$. In this case, $\mu$ may admit a density with respect to an $m$-dimensional reference measure (e.g. the $m$-dimensional Hausdorff measure $\mathcal{H}^m$), but critically, *no density exists with respect to Lebesgue measure in the ambient space.* Such situations appear frequently in scientific domains where physical constraints restrict the degrees of freedom of the data.

To clarify the kinds of manifolds we study here, we briefly introduce some definitions from (Aamari et al., 2019).

**Definition 4.1** (Medial Axis). The *medial axis*, denoted as $\Sigma(\mathcal{M})$, is the set of points $x \in \mathbb{R}^d$ where the nearest-point projection is multi-valued.

**Definition 4.2** (Reach). The *reach* of a manifold $\mathcal{M}$ is the minimal distance from any point on $\mathcal{M}$ to $\Sigma(\mathcal{M})$.

**Assumptions 4.3.** *Let* $\mathcal{M} : \{x \in \mathbb{R}^d : h(x) = 0\}$, *where* $h$ *is analytically known. Assume* $\mathcal{M}$ *to be* ***compact,*** $\mathcal{C}^2$ ***(twice-differentiable), and closed with positive reach,*** *with* $p_0$ *lying on* $\mathcal{M}$ *and* ***varying in all tangential directions*** *(to ensure that* $p_0$ *does not live on a proper submanifold of* $\mathcal{M}$ *itself). Also, require that* $\mathcal{M}$ *has full-rank Jacobian matrices.*

For all theorems and corollaries in this paper, the assumptions in 4.3 hold. In particular, $p_0$ is a density with respect to the $m$-dimensional Hausdorff measure $\mathcal{H}^m$, with law $\mu_0$ where $d\mu_0 = p_0$. We write the codimension of $\mathcal{M}$ as $k = d - m$. As a consequence of the Implicit Function Theorem, Jacobians may be obtained at any point $x \in \mathcal{M}$, enabling identification of normal vectors to the manifold.

**Theorem 4.4** (Geometric Non-degeneracy of $p_\sigma$). *The new distribution* $p_\sigma$ *is not strictly supported on a lower-dimensional manifold.*

*Proof.* Adding Gaussian noise in $N_z\mathcal{M}$ at each $z \in \mathcal{M}$ thickens $p_0$ into a tubular distribution centered around $\mathcal{M}$ whose support contains all affine normal fibers $z + N_z\mathcal{M}$, yielding $d$-dimensional support. $\square$

We now discuss the law of the projected samples.

**Definition 4.5** (Distribution of the Projected Samples). Define the pushforward of $p_\sigma$ under $\Pi$ as $\Pi_{\#}p_\sigma$. We will often refer to $\Pi_{\#}p_\sigma$ as $\tilde{p}_\sigma$ for conciseness.

The nearest-point projection $\text{Proj}_{\mathcal{M}}(x)$ is unique if $x$ lies within the *reach tube*, or the tube around $\mathcal{M}$ with the reach as its radius. Our bounds for nonlinear $\mathcal{M}$ are executed entirely within such tubes and simply bound possible contributions from outside the tubes, and therefore the choice of tie-breaking in the selection of $\Pi$ is arbitrary.

We first prove that $\Pi_{\#}p_\sigma$ and $p_0$ are exactly identical when $h$ is linear (i.e. when $\mathcal{M}$ has no curvature).

**Theorem 4.6** (Perfect Recovery under Linear Constraints). *Let* $h : \mathbb{R}^d \rightarrow \mathbb{R}^{d-m}$ *be linear, e.g.* $h(x) = Ax - b, A \in \mathbb{R}^{m \times d}, b \in \mathbb{R}^m$. *Then,* $p_0(x) = \tilde{p}_\sigma(x)$ *for all* $x \in \mathcal{M}$.

*Proof.* Recall that $p_\sigma$ is generated by adding noise strictly in $N_{x'}\mathcal{M}$ to samples $x'$ to obtain ambient samples $y$. For linear $\mathcal{M}$, the nearest point from such $y$ to $\mathcal{M}$ is $x'$ itself. $\square$

We remark that this is already immediately an advantage of our proposed approach compared to existing isotropic techniques: at any nonzero noising scale, isotropic noising followed by post-projection *immediately* induces distortion, as is seen in the linear manifold example in Figure 1.

We now bound the total variation (TV) for general nonlinear $\mathcal{M}$ using the (intrinsic) TV as it is defined for probability densities with the $\mathcal{H}^m$ measure on $\mathcal{M}$ (Gibbs & Su, 2002):

$$\text{TV}(p, q) = \frac{1}{2}\int_{\mathcal{M}} |p(x) - q(x)| d\mathcal{H}^m(x).$$

**Theorem 4.7** (Total Variation Bound). *The total variation distance (on* $\mathcal{M}$, *in* $\mathcal{H}^m$*) between* $\tilde{p}_\sigma$ *and* $p_0$ *is bounded as*

$$\text{TV}(\tilde{p}_\sigma, p_0) \leq C_1 e^{\frac{-C_2 r^2}{\sigma^2}},$$

*with* $r < \text{reach}(\mathcal{M})$ *and constants depending only on* $k$.

*Proof.* See Appendix B. The idea is to construct a coupling between $\tilde{p}_\sigma$ and $p_0$. $\square$

**Corollary 4.8** (Local Reach Bound). *Define* $B_\rho(z) = \{z + n : \|n\| < \rho\}$. *Let* $\tau(\cdot)$ *denote the* pointwise reach *function*

$$\tau(z) := \sup\{\rho > 0 : B_\rho(z) \text{ uniquely projects to } z\}.$$

*Then, for each* $z \in \mathcal{M}$,

$$D_\sigma(z) := \mathbb{E}\left[\text{dist}\left(z, \Pi(z+n)\right)\right] \leq C_1 e^{\frac{-C_2 \tau(z)^2}{\sigma^2}}$$

*and consequently*

$$\text{TV}(\tilde{p}_\sigma, p_0) \leq 2L\sigma C_1 \int_{\mathcal{M}} e^{-C_2 \frac{\tau(z)^2}{\sigma^2}} d\mathcal{H}^m(z).$$

*Proof.* See Appendix B. The proof is similar to that of Theorem 4.7, but with a less restrictive global reach tube. $\square$

We also remark that Theorem 4.6 can also be understood as an immediate consequence of Theorem 4.7, as the reach of a linear manifold is infinity.

In practice, one may not need to worry about high curvature regions if probability mass is concentrated in low curvature regions. We find that $\sigma < \text{reach}(\mathcal{M})$ is a safe initial recommendation. We remark that the manifold reach is inversely proportional to its curvature, with (Aamari et al., 2019) being a good reference for further details. Ultimately, our theoretical analysis indicates that for applications where either (a) manifold global curvature is minimal, inducing high global or local reach, or (b) probability mass is estimated to be concentrated in regions of low local curvature, one can anticipate little distortion from learning and projecting $p_\sigma$.

Given the above results, one may reliably train any generative model (DDPM, SBDM, NF, etc.) to model $p_\sigma$, not $p_0$, and project generated samples knowing that this procedure induces little to zero error. One can anticipate that generative modeling paradigms that operate in the manner of (1) will greatly benefit from our proposed approach, which directly targets the issue of dimension mismatch between the supports of the latent and the target distributions. Our experiments show both this minimal distortion as well as the expected improved sampling stability.

# 5. Numerical Experiments

We present experiments in order of ascending task complexity and scientific relevance. We first study distributions of 3-D points on 2-D manifolds (a plane, sphere, and mesh surface), followed by a 784-D image task and a 90-D application in protein backbone generation. Experimental configurations are delineated in Appendix C, and evaluation metrics are precisely described in Appendix D. For diffusion models, we implement PIDM, PDM, and DDPM on $p_0$, with DDPM approaches also being used to model $p_\sigma$ and an isotropically noised $p_0$ with the standard deviation matching the $\sigma$ of the corresponding $p_\sigma$. We present NF results on the first two tasks to demonstrate reduced Jacobian determinant magnitude, with $p_\sigma$, $p_0$, and the isotropically noised $p_0$. We experiment with both RealNVP (Dinh et al., 2017) and Glow (Kingma & Dhariwal, 2018) architectures to illustrate generalizability across multiple choices of NF parametrizations. We report sampling time in seconds and training time in seconds per batch.

## 5.1. Plane

We first discuss a simple distribution of 3-D points supported strictly on a 2-D plane. Precisely, samples $x$ lie on

$$\mathcal{M}_{\text{plane}} := \{x \in \mathbb{R}^3 : Ax = b\},$$

where in our case, $A = [1, 2, 3]$ and $b = 1$.

Due to computationally equivalent training objectives, it is expected that training time for $p_\sigma$, DDPM, projected DDPM (trained identically to DDPM), and PDM are close to identical. Table 1 includes the same metrics computed both extrinsically (all methods) and intrinsically (if applicable), where the latter is included for alignment with Section 4. We see that learning $p_\sigma$ always leads to optimal or competitive distribution quality compared to other techniques. Figure 2 confirms this is consistent across various $\sigma$, Figure 3 illustrates the qualitative improvement offered by our approach, and Figure 4 shows that learning $p_\sigma$ indeed offers the expected benefits of reduced score and log-determinant magnitude.

## 5.2. Sphere

We then study the same distribution on a sphere, introducing global curvature while maintaining similar task complexity:

$$\mathcal{M}_{\text{sphere}} = \{x \in \mathbb{R}^3 : \|x\|_2 = 1\} \quad (2)$$

| | | Method | Train time | Sampling time | COV ↑ | JSD ↓ | TVD ↓ |
|---|---|---|---|---|---|---|---|
| **Plane** | **DMs** | PDM | 0.0012 | 0.4991 | 0.8678 | 0.0555 | 0.2270 |
| | | PIDM | 0.0019 | 0.1959 | 0.8119 | 0.0873 | 0.2913 |
| | | $\tilde{p}_\sigma$ | 0.0012 | 0.2908 | 0.8736 | 0.0515 | 0.2015 |
| | | DDPM | 0.0012 | 0.1960 | 0.8632 | 0.0475 | 0.1967 |
| | | DDPM (proj.) | 0.0012 | 0.1966 | **0.8852** | **0.0442** | **0.1893** |
| | | DDPM (proj., iso.) | 0.0012 | 0.1966 | 0.8096 | 0.0821 | 0.2817 |
| | **NFs** | Glow (iso.) | 0.0169 | 0.6182 | 0.9358 | **0.1026** | **0.3135** |
| | | Glow ($\tilde{p}_\sigma$, ours) | 0.0169 | 0.6181 | 0.9219 | 0.1056 | 0.3187 |
| | | Glow (proj.) | 0.0170 | 0.6197 | **0.9115** | 0.2826 | 0.5962 |
| | | Glow | 0.0170 | 0.6176 | **0.9115** | 0.2790 | 0.5931 |
| | | RealNVP (iso.) | 0.0062 | 0.1191 | 0.9340 | 0.0506 | **0.2034** |
| | | RealNVP ($\tilde{p}_\sigma$, ours) | 0.0062 | 0.1193 | **0.9115** | **0.0500** | 0.2038 |
| | | RealNVP (proj.) | 0.0055 | 0.1193 | 0.9392 | 0.1433 | 0.3853 |
| | | RealNVP | 0.0055 | 0.1188 | 0.9358 | 0.1417 | 0.3880 |
| **Sphere** | **DMs** | PDM | 0.0012 | 0.3763 | 0.8888 | 0.0771 | 0.2733 |
| | | PIDM | 0.0020 | 0.1965 | 0.5664 | 0.1570 | 0.4058 |
| | | $\tilde{p}_\sigma$ | 0.0012 | 0.2656 | **0.9105** | **0.0657** | **0.2341** |
| | | DDPM | 0.0012 | 0.1975 | 0.7976 | 0.0777 | 0.2694 |
| | | DDPM (proj.) | 0.0012 | 0.1979 | 0.9011 | 0.0662 | 0.2393 |
| | | DDPM (proj., iso.) | 0.0012 | 0.1979 | 0.8717 | 0.0989 | 0.3097 |
| | **NFs** | Glow | 0.0161 | 0.3927 | 0.7685 | 0.3475 | 0.6708 |
| | | Glow ($\tilde{p}_\sigma$, ours) | 0.0161 | 0.3583 | 0.8837 | **0.0911** | **0.2925** |
| | | Glow (proj.) | 0.0161 | 0.3938 | 0.8686 | 0.0963 | 0.3000 |
| | | Glow (iso.) | 0.0161 | 0.3938 | **0.9181** | 0.1360 | 0.3727 |
| | | RealNVP | 0.0055 | 0.0660 | 0.8018 | 0.3389 | 0.6672 |
| | | RealNVP ($\tilde{p}_\sigma$, ours) | 0.0055 | 0.0710 | **0.8675** | **0.0769** | **0.2638** |
| | | RealNVP (proj.) | 0.0055 | 0.0664 | 0.7048 | 0.2817 | 0.5969 |
| | | RealNVP (iso.) | 0.0055 | 0.0664 | 0.7108 | 0.3114 | 0.6295 |

Extrinsic metrics.

| | | Method | COV ↑ | JSD ↓ | TVD ↓ |
|---|---|---|---|---|---|
| **Plane** | **DMs** | PDM | 0.8678 | 0.0438 | 0.1999 |
| | | $\tilde{p}_\sigma$ | 0.8736 | 0.0400 | 0.1810 |
| | | DDPM (proj.) | **0.8852** | **0.0340** | **0.1667** |
| | | DDPM (proj., iso.) | 0.8096 | 0.0686 | 0.2543 |
| | **NFs** | Glow (iso.) | 0.8539 | **0.0109** | **0.0999** |
| | | Glow ($\tilde{p}_\sigma$, ours) | **0.8560** | 0.0114 | 0.1049 |
| | | Glow (proj.) | 0.6393 | 0.0362 | 0.2059 |
| | | RealNVP (iso.) | 0.9062 | **0.0035** | **0.0607** |
| | | RealNVP ($\tilde{p}_\sigma$, ours) | **0.9182** | 0.0040 | 0.0623 |
| | | RealNVP (proj.) | 0.8105 | 0.0206 | 0.1660 |
| **Sphere** | **DMs** | PDM | 0.8883 | 0.0489 | 0.2223 |
| | | $\tilde{p}_\sigma$ | **0.9105** | **0.0369** | **0.1791** |
| | | DDPM (proj.) | 0.8985 | 0.0378 | 0.1841 |
| | | DDPM (proj., iso.) | 0.8724 | 0.0684 | 0.2637 |
| | **NFs** | Glow ($\tilde{p}_\sigma$, ours) | 0.8792 | **0.0581** | **0.2264** |
| | | Glow (proj.) | **0.8860** | 0.0600 | 0.2271 |
| | | Glow (iso.) | 0.8480 | 0.0975 | 0.3193 |
| | | RealNVP ($\tilde{p}_\sigma$, ours) | **0.8958** | **0.0516** | **0.2156** |
| | | RealNVP (proj.) | 0.6986 | 0.2254 | 0.5381 |
| | | RealNVP (iso.) | 0.6579 | 0.2700 | 0.5898 |

Intrinsic metrics (using 2-D representations).

*Table 1.* Metrics for plane and sphere tasks at $\sigma = 0.05$. Among these experiments, learning $p_\sigma$ always either improves the sampled distribution or is competitive compared to either learning $p_0$ or modifying the learning or sampling algorithms for the diffusion models. For NF approaches, we highlight the best-performing method within each architecture.

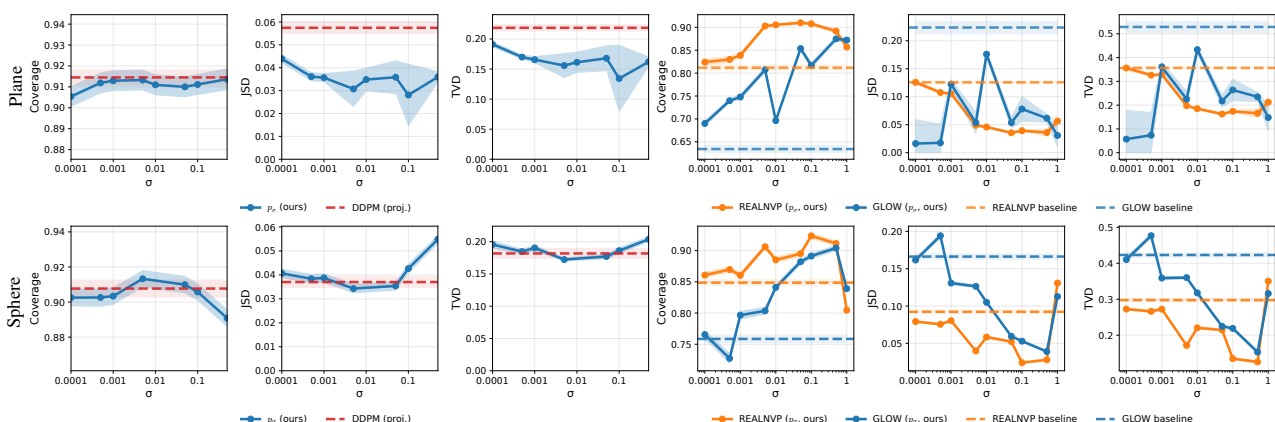

*Figure 2.* Metrics across varied $\sigma$ for diffusion model and NF approaches on toy tasks. In all cases, learning $p_\sigma$ consistently outperforms or is competitive against learning $p_0$ and post-projecting samples, with the expected possible performance decrease as $\sigma \to \mathrm{reach}(\mathcal{M})$.

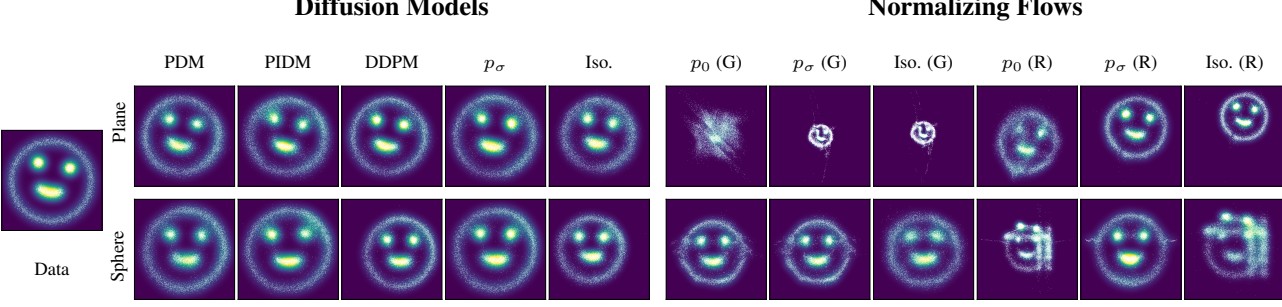

*Figure 3.* Image examples for plane and sphere tasks. In all cases, $p_\sigma$ samples consistently visually outperform or are competitive against the other techniques. The label "(G)" refers to Glow and the label "(R)" refers to RealNVP.

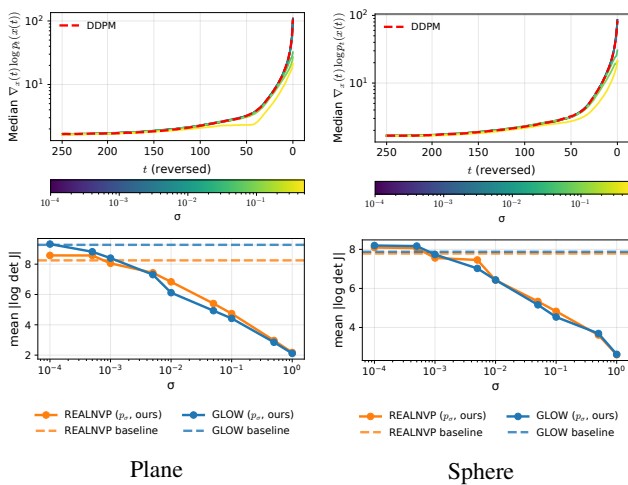

*Figure 4.* Sampling stability for plane and sphere tasks. We observe the expected reduction in score and Jacobian log-determinant magnitude across both tasks and generative modeling paradigms. We present similar results for the complex tasks in Appendix E.

Table 1 shows that $p_\sigma$ yields improved results across all metrics and is robust in providing competitive or improved performance within each paradigm. This is visually clear

in Figure 3. We also remark that in Figure 2, performance worsens as $\sigma$ approaches 1.0. This aligns with results in Section 4, as $\mathrm{reach}(\mathcal{M}_{\mathrm{sphere}}) = 1.0$. This is notably not as dramatic for the plane task, as $\mathrm{reach}(\mathcal{M}_{\mathrm{plane}}) = \infty$, and any performance decrease is likely due to worsening data normalization with too-large $\sigma$.

### 5.3. Mesh

Here, we present results for a standard Gaussian supported strictly on the surface of the Stanford Bunny mesh (Stanford University Computer Graphics Laboratory). This experiment is inspired by results in (Elhag et al., 2023). We choose this task as it maintains the same dimensionalities of the tasks in Sections 5.1-5.2 while introducing greater and more irregular local curvature.

In Figure 5, we see that the DDPM and PIDM approaches frequently produce off-manifold samples. Although projected DDPM samples visually align well with $p_0(x)$, they consistently violate constraints prior to projection. This suggests that $p_\sigma$ may be more reliable when both distributional fidelity and adherence to constraints are critical, as is evidenced by both these off-manifold samples as well as the

competitive JSD and TVD metrics in Table 2. Due to the challenge of intrinsic metrics on such a complicated mesh, we use ambient space histogramming in $\mathbb{R}^d$ for the JSD and TVD metrics.

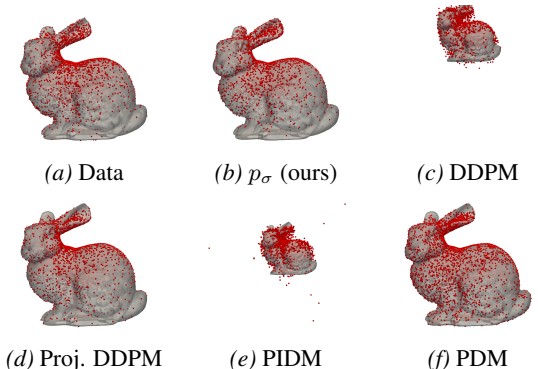

*(a)* Data     *(b)* $p_\sigma$ (ours)     *(c)* DDPM

*(d)* Proj. DDPM     *(e)* PIDM     *(f)* PDM

*Figure 5.* Mesh task samples. The projected $p_\sigma$ samples are close to $p_0$ and do not risk off-manifold samples as others do.

| Method | Train time | Sampling time | COV ↑ | JSD ↓ | TVD ↓ |
|---|---|---|---|---|---|
| PDM | 0.0013 | 8.4991 | 0.8626 | 0.1912 | 0.4318 |
| PIDM | 0.0033 | 0.2196 | 0.5952 | 0.1298 | 0.3248 |
| $p_\sigma$ (ours) | 0.0012 | 0.3663 | 0.8506 | 0.1484 | 0.3471 |
| DDPM | 0.0013 | 0.2246 | 0.7104 | **0.1266** | **0.3090** |
| DDPM (proj.) | 0.0013 | 0.2415 | **0.8643** | 0.1480 | 0.3458 |

*Table 2.* Mesh task metrics at $\sigma = 0.0005$. Learning $p_\sigma$ is consistently competitive with other methods while avoiding outliers and off-manifold samples through projection.

## 5.4. Image Generation with Total Flux Constraint

It is often the case in scientific inverse imaging tasks that the total flux of an image, or the sum of all pixel intensities, is constrained (Feng et al., 2025; The EHT Collaboration, 2019). To that end, we present a task of generating images with fixed total flux, where the sum of all pixel intensities is 100.0. We implement this task using $28 \times 28$ MNIST images (Deng, 2012), which may be represented as vectors $x \in \mathbb{R}^{784}$. This example is presented for its clear visual notion of sample quality. The constraint manifold is thus

$$\mathcal{M}_{\text{flux}} := \left\{ x : \sum_{i=1}^{784} x_i = 100 \right\}.$$

Due to the challenge of histogram-based metrics in higher dimensions, we use learned embedding-based metrics inspired by the popular Frechét Inception Distance (Heusel et al., 2017), as well as an empirical JSD over digit class distributions as inferred over a trained classifier. The precise procedure used to obtain these distances is described in thorough detail in Appendix D.

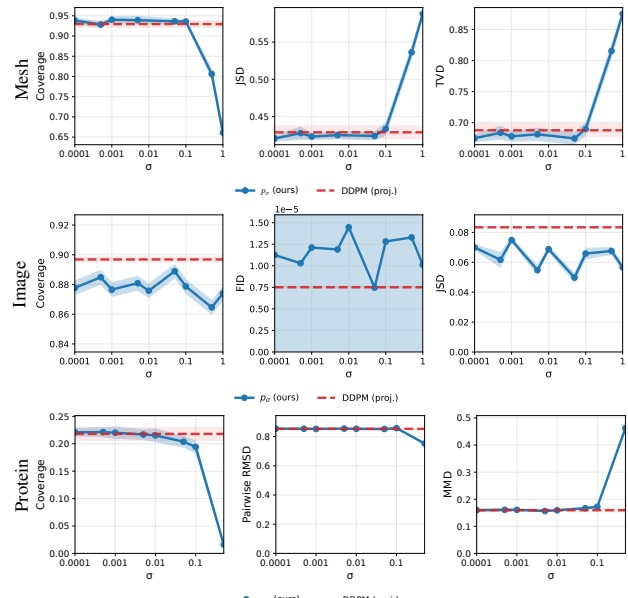

*Figure 6.* Metrics across varied $\sigma$ on complex tasks. Learning $p_\sigma$ consistently improves upon the projected DDPM baseline on all advanced problems. We remark that the coverage and FID metrics are computed in the embedding space due to the challenges of distributional evaluation of images, and their competitive performance combined with consistently outperforming class JSD illustrates the utility of our approach for constrained image generation.

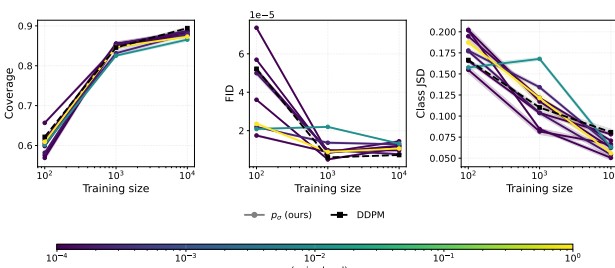

*Figure 7.* Performance improvement with respect to number of samples and $\sigma$ on MNIST task. Learning $p_\sigma$ at the tested $\sigma$ levels consistently improves performance on all metrics, with an expected increase as the number of training samples increases.

| Method | Train time | Sampling time | COV ↑ | FID ↓ | Class JSD ↓ |
|---|---|---|---|---|---|
| $p_\sigma$ (ours) | 0.0083 | 39.0945 | 0.8765 | $1.512 \times 10^{-5}$ | **0.0730** |
| PDM | 0.0083 | 78.1216 | 0.8457 | $2.218 \times 10^{-5}$ | 0.0874 |
| DDPM | 0.0083 | 39.0982 | **0.8954** | $7.434 \times 10^{-6}$ | 0.0868 |
| PIDM | 0.0089 | 39.0990 | 0.4713 | $1.665 \times 10^{-4}$ | 0.2174 |
| DDPM (proj.) | 0.0083 | 39.1013 | 0.8954 | $\mathbf{7.379 \times 10^{-6}}$ | 0.0869 |

*Table 3.* MNIST metrics at $\sigma = 0.01$ with 10,000 training samples.

In ablations against $\sigma$ and the total training dataset size presented simultaneously in Figure 7, we see that our approach improves or provides competitive performance on the MNIST task on all dataset sizes and choices of $\sigma$. We

also remark that such results are consistent for fixed 10K training samples in Figure 6, wherein learning $p_\sigma$ is uniformly close to or outperforms the projected DDPM baseline on all metrics.

### 5.5. Protein Backbone Generation

To demonstrate the effectiveness of our approach on real scientific data, we apply our technique to the generation of protein backbone fragments, which are the repeated N-CA-C atom structures present at the core of protein samples. Using generative AI to model the distribution of realistic protein backbones is an active area of research (Wu et al., 2024; Zhang et al., 2024; Xu et al., 2021), with downstream utility for prior-building in *de novo* and inverse design of proteins.

Protein backbone fragments abide by several underlying physical constraints: in particular, bond angles and lengths are rigidly constrained due to known chemical properties, imposing a geometric constraint on how the structures appear in 3-D space. Thus, applying out-of-the-box generative modeling techniques bears the risk of generating geometrically implausible structures. Moreover, existing state-of-the-art approaches to constraint-aware molecular structure generation, such as torsional diffusion (Jing et al., 2022), specifically cite protein generation, particularly in the backbone fragments, to be challenging due to the dimensionality of proteins. We do not claim to achieve SOTA performance against the many highly-tailored generative models constructed for computational biology applications. Rather, we present this as a highly realistic testbed for our approach with a complex constraint manifold and large data dimensionality, illustrating the potential of our technique for large-scale scientific generative modeling problems.

A single sample (a *fragment*) has size (# residues, # atoms, # coordinates), where the second dimension indexes the atom (N, CA, C) and the third dimension indexes the coordinates $(x, y, z)$ in 3-D space for that atom. For each bond, we have a *bond length* and a *bond angle* constraint (the latter of which is taken over triplets of bonded atoms).

Let the coordinates of atom $a$ in residue $i$ be denoted by $x_{i,a} \in \mathbb{R}^3$. The full coordinate vector is then $x \in \mathbb{R}^D$ where $D = 9L$ (three atoms per residue, each with three coordinates, where $L$ is the number of residues in the fragment). We use fragments with lengths of 10 residues extracted from the CASP12 dataset as presented in SideChainNet (King & Koes, 2021; 2024), a dataset expanding the SOTA ProteinNet (AlQuraishi, 2019) with full backbone geometry information. We train on a subset of 20K fragments. Metrics are evaluated on sets of 1K samples.

In Table 4, we see that our approach is competitive against other approaches on coverage and MMD with a bandwidth

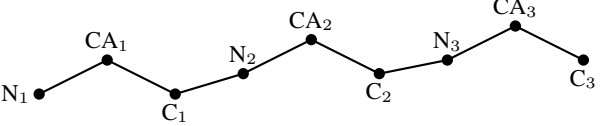

*Figure 8.* Backbone fragment with atom coordinates as generated variables. In our setting, a single data sample consists of $L$ residues ($L = 3$ in this figure) with each residue made up of 3 backbone atoms (N, CA, C), each represented by $(x, y, z)$-coordinates.

| Method | Train time | Sampling time | COV | Pairwise RMSD | MMD |
|---|---|---|---|---|---|
| $p_\sigma$ (ours) | 0.0040 | 5.5825 | 0.2200 | **1.6262** | $3.37 \times 10^{-4}$ |
| DDPM | 0.0041 | 1.6013 | 0.2180 | 0.7414 | $\mathbf{3.35 \times 10^{-4}}$ |
| PDM | 0.0041 | 270.3082 | **0.3630** | 1.3534 | $3.80 \times 10^{-4}$ |
| DDPM (proj.) | 0.0041 | 6.8878 | 0.2880 | 0.8445 | $3.36 \times 10^{-4}$ |
| PIDM | 0.0070 | 1.6486 | 0.2240 | 1.4216 | $3.37 \times 10^{-4}$ |

*Table 4.* Protein backbone fragment metrics at $\sigma = 0.001$.

of 1.0. Moreover, as indicated by the Pairwise RMSD score (a metric of diversity amongst samples), this is accomplished without collapse to memorization of a single sample. While DDPM performs slightly better on the MMD metric, it violates constraints. In comparison, the $p_\sigma$ samples perform well on metrics of distributional fidelity and diversity with the added benefit of guaranteed constraint adherence (up to numerical optimization challenges) by construction of our approach. This, combined with the reduction in score magnitudes (which we empirically show in Appendix E), indicates that our modified distribution may be strongly suitable for biological and molecular applications where strict geometric constraints are present.

## 6. Limitations

We strictly consider the case where $\mathcal{M}$ is known analytically, which is the case in a broad range of applications in science where physical laws are known precisely from domain knowledge. However, there is still potential utility for our proposed technique in cases where the manifold is known only approximately. In particular, precise manifold knowledge is less important than knowledge of normal directions to $\mathcal{M}$, which is all that is required for the first stage of our approach. Thus, manifold approximation are acceptable so long as local curvature is accurate.

Our theoretical results require that $\mathcal{M}$ is $\mathcal{C}^2$, which may be mildly restrictive for certain constraints. In practice, some gentle smoothing on non-smooth regions (e.g. mesh edges) may be needed if probability mass is present at those points. However, the only requirement is the ability to compute local Jacobians on $\mathcal{M}$ to extract normal vectors (where we assume Jacobians are full-rank at each $x \in \mathcal{M}$), so regularity may be relaxed from $\mathcal{C}^2$ to $\mathcal{C}^1$ in implementation. In fact, for mere implementation and not theoretical justification,

$\mathcal{M}$ need only be differentiable where $p_\sigma$ is nonzero. An additional potential theoretical limitation is our omission of possible training error in the model approximations of either $p_0$ or $p_\sigma$, which may induce a gap between theory and practice.

We also note that while our method introduces only a single new hyperparameter, it is true that it may require some amount of tuning for particularly complex manifolds. However, while too-large $\sigma$ can be harmful for manifolds of small global reach (or distributions concentrated in regions of small local reach), our results generally show that extremely small $\sigma$ yielded either similar or often improved performance compared to the raw (or raw and post-projected) baselines. For users constrained by computational budgets, we recommend overly conservative $\sigma$ for initial experiments with gradual increases if necessary for improved results in order to optimally implement our method.

Finally, we note that the computational needs of orthogonal noising and postprojection can be large in some complex settings. This may yield nontrivial computational cost in certain settings.

## 7. Conclusions and Future Work

In this work, we demonstrated how a strategic pre- and post-processing step, in which we modify the underlying distribution being learned by a generative model, can circumvent common pitfalls in the generative modeling of manifold-supported distributions. Our technique offers automatic guarantees of both manifold adherence in samples through post-projection, which is not true for other approaches of similar computational requirements, as well as accurate distribution recovery according to intrinsic metrics on the manifold. We also present theoretical and experimental results which demonstrate that our approach either improves upon or is competitive against SOTA generative modeling techniques while also frequently offering drastically reduced training or sampling time compared to other tailored constraint-aware approaches. We also observed in all experiments that our approach either outperforms other approaches according to evaluation metrics or performs competitively with drastically reduced sampling time, a known bottleneck in existing applications of generative models. Our examples on various data structures and task complexity demonstrate the utility of our approach in a wide variety of practical and scientific settings.

In this work, we experiment with learning $p_\sigma$ using multiple popular generative modeling techniques, specifically those falling under diffusion model and NF paradigms. However, our approach is highly flexible and not restricted to these training parameterizations; our results support the claim that learning and projecting $p_\sigma$ is a much more stable and reliable way to apply generative models to constrained tasks regardless of the chosen technique for distribution modeling.

One can also consider our approach applied to diffusion models from an SDE perspective. So far, we have essentially modified the forward process to apply 1. anisotropic noise first to push points off $\mathcal{M}$ and then 2. the usual isotropic noising to allow standard training. One could extend our approach to replace the $\beta(t)$ noise variance schedule in the diffusion model forward process with a spatially-dependent $\beta(x, t)$ (or perhaps $\beta(x, t, \mathcal{M})$). Future work may consider a continuous interpolation between isotropic and our anisotropic noising in the diffusion model SDEs.

Moreover, alternative noising strategies could also be investigated. We only consider Gaussian noising in the normal bundle as it is straightforward and enables a fair comparison between many existing works in the literature. However, one could consider alternative distributions with a mean of zero with bounded norm less than the reach to enable improved distributional recovery.

Our theoretical analyses demonstrate that the error between $p_\sigma(x)$ and $p_0(x)$ is dominated by manifold curvature via the reach. Future work may investigate defining a *spatially-dependent* $\sigma_x$, which can be larger in linear regions and reduced in regions of greater, more irregular curvature. Expanding $p_\sigma$ to be aware of second-order manifold information through curvature is a natural next step for combating challenges in sampling from degenerate distributions.

## Acknowledgements

This material is based upon work supported by the U.S. Department of Energy, Office of Science, Office of Advanced Scientific Computing Research, Department of Energy Computational Science Graduate Fellowship under Award Number(s) DE-SC0023112. The authors also acknowledge support from NSF award DMS 2038118.

## Impact Statement

This paper presents work with the broader goal of aligning AI for domain science use cases, where ensuring mathematical guarantees and adherence with known constraints is particularly crucial. In many of these applications, ensuring synergy between known laws originating from domain sciences and data-driven techniques in generative AI is crucial for meaningful implementation in actual science workflows. As such, the potential societal consequences of our work match those of general AI4Science endeavors.

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

# A. Notation Glossary

| Symbol | Description |
|---|---|
| *Spaces and Manifolds* | |
| $\mathbb{R}^d$ | Ambient space (dimension $d$) |
| $\mathcal{M}$ | $m$-dimensional constraint manifold embedded in $\mathbb{R}^d$ |
| $m$ | Intrinsic dimension of $\mathcal{M}$ |
| $k = d - m$ | Codimension of $\mathcal{M}$ |
| $T_x\mathcal{M}$ | Tangent space to $\mathcal{M}$ at point $x$ |
| $N_x\mathcal{M}$ | Normal space to $\mathcal{M}$ at point $x$ |
| $\mathrm{reach}(\mathcal{M})$ | Global reach of manifold $\mathcal{M}$ |
| $\tau(z)$ | Pointwise reach function at $z \in \mathcal{M}$ |
| $r < \mathrm{reach}(\mathcal{M})$ | Radius of reach tube $T_r(\mathcal{M})$ |
| *Constraints* | |
| $h : \mathbb{R}^d \to \mathbb{R}^m$ | Constraint function defining $\mathcal{M} = \{x : h(x) = 0\}$ |
| $J_h(x)$ | Jacobian of constraint function at $x$ |
| *Distributions and Measures* | |
| $p_0(x)$ | Original data distribution on $\mathcal{M}$ |
| $p_\sigma(x)$ | Perturbed (lifted) distribution in $\mathbb{R}^d$ |
| $\tilde{p}_\sigma(x)$ | Projected distribution $\Pi_\# p_\sigma$ on $\mathcal{M}$ |
| $p_T(x)$ | Terminal/latent distribution (typically $\mathcal{N}(0, I)$) |
| $\mu$ | Probability measure/law |
| $\lambda^d$ | $d$-dimensional Lebesgue measure |
| $\mathcal{H}^m$ | $m$-dimensional Hausdorff measure |
| *Random Variables* | |
| $Z$ | Random variable with law $p_0$ on $\mathcal{M}$ |
| $N\|Z = z$ | Conditional Gaussian noise in $N_z\mathcal{M}$ |
| $X = Z + N$ | Lifted random variable with law $p_\sigma$ |
| $Y = \Pi(X)$ | Projected random variable with law $\tilde{p}_\sigma$ |
| *Operators and Functions* | |
| $\Pi : \mathbb{R}^d \to \mathcal{M}$ | Nearest-point projection onto $\mathcal{M}$ |
| *Hyperparameters* | |
| $\sigma$ | Noise scale for perturbation |
| $t \in [0, T]$ | Virtual time for diffusion process |
| $T$ | Terminal time (typically $T = 250$ in experiments) |
| *Metrics* | |
| $\mathrm{TV}(p, q)$ | Total variation distance: $\frac{1}{2}\int_\mathcal{M}|p(x) - q(x)|d\mathcal{H}^m(x)$ |
| $\mathrm{JSD}(p\|q)$ | Jensen-Shannon divergence |
| COV | Coverage metric |
| FID | Fréchet Inception Distance. |
| RMSD | Root mean square deviation |
| MMD | Maximum Mean Discrepancy |

*Table 5.* Summary of notation used throughout this paper.

We note that while we use the term "FID" to refer to the popular metric for image generation analysis, our implementation is an approximation using a LeNet classifier instead of the Inception-V3 architecture standard for FID. The precise details of our implementation are described in Appendix D.

## B. Theory

**Lemma B.1** ($\mathcal{C}^1$-ness of $\Pi$ on $\mathbb{R}^d \setminus \Sigma(\mathcal{M})$). *If $\mathcal{M} \in \mathcal{C}^2$, then $\Pi(x) : \mathbb{R}^d \setminus \Sigma(\mathcal{M}) \to \mathcal{M} \in \mathcal{C}^1$. Moreover, as $\mathcal{C}^1$ implies being locally Lipschitz, $\Pi$ is locally Lipschitz on this domain.*

*Proof.* This is a direct consequence of the main result in (Leobacher & Steinicke, 2021), which states that for $\mathcal{C}^k$-submanifolds with $k \geq 2$, $\Pi$ is $\mathcal{C}^{k-1}$ on the maximal open domain where $\Pi$ is uniquely valued. $\qquad\square$

**Theorem B.2** (Difference of $\mathrm{Proj}_{\mathcal{M}}$ and $\Pi$). *Under the assumptions of $\mathcal{M}$ in this paper, $\lambda^d (\Sigma(\mathcal{M})) = 0$.*

*Proof.* By (Chazal et al., 2010), $\Sigma(\mathcal{M})$ has $\mathcal{H}^d$-measure zero, and as $\mathcal{H}^d$ and $\lambda^d$ agree by Chapter 2.2 in (Evans & Gariepy, 2015), $\lambda^d (\Sigma(\mathcal{M})) = 0$. $\qquad\square$

**Lemma B.3** (Gaussian Tails of $p_\sigma$ Outside of $T_r(\mathcal{M})$). *We have*

$$\int_{\mathbb{R}^d \setminus T_r(\mathcal{M})} p_\sigma(x)dx \leq C_1 e^{-C_2 \frac{r^2}{\sigma^2}}, C_1 = 2^{\frac{(d-m)}{2}}, C_2 = \frac{1}{4}.$$

*Proof.* We have

$$\int_{\mathbb{R}^d \setminus T_r(\mathcal{M})} p_\sigma(x)dx = \mathbb{P}\left(X \in \mathbb{R}^d \setminus T_r(\mathcal{M})\right).$$

Recall that the procedure for forming $p_\sigma$ yields $X = Z + N$ with $N$ conditional on $Z = z$ being sampled from $\mathcal{N}(0, \sigma^2 I_{N_z \mathcal{M}})$, note that for any $(Z, N) = (z, n)$, we have

$$z + n \in T_r(\mathcal{M}) \Leftrightarrow \|n\| < r.$$

Thus, for any particular $X = Z + N$, we have

$$\{X \notin T_r(\mathcal{M})\} = \{\|N\| \geq r\}.$$

Therefore, the remainder of the proof amounts to simply identifying an upper bound on the tail probability of the $k$-dimensional Gaussian $N$.

It is known that the squared norm of $N$ follows a $\chi^2$ distribution with $k$ degrees of freedom:

$$\frac{\|N\|^2}{\sigma^2} \sim \chi_k^2.$$

This gives

$$\mathbb{P}(\|N\| \geq r) = \mathbb{P}\left(\frac{\|N\|^2}{\sigma^2} \geq \frac{r^2}{\sigma^2}\right) = \mathbb{P}(U \geq u),$$

in which $U \sim \chi_k^2$ and $u := \frac{r^2}{\sigma^2}$.

The moment generating function for a random variable following a $\chi_k^2$ distribution is

$$\mathbb{E}\left[e^{\lambda U}\right] = (1 - 2\lambda)^{-\frac{k}{2}}.$$

Let $\lambda = \frac{1}{4} \in (0, \frac{1}{2})$. Then

$$(1 - 2\lambda)^{-\frac{k}{2}} = \left(1 - \frac{1}{2}\right)^{-\frac{k}{2}} = 2^{\frac{k}{2}}$$

and

$$\mathbb{P}(U \geq u) = \mathbb{P}(e^{\lambda U} \geq e^{\lambda u})$$
$$\leq e^{-\lambda u} \mathbb{E}\left[e^{\lambda U}\right] \text{ by Markov's inequality}$$
$$\leq e^{-\frac{1}{4}u} 2^{\frac{k}{2}}$$
$$= 2^{\frac{k}{2}} e^{-\frac{1}{4}\frac{r^2}{\sigma^2}}.$$

$\square$

*Proof of Theorem 4.7.* Recall that $p_0$ is a density on $(\mathcal{M}, \mathcal{H}^m)$, and that $p_\sigma$ is obtained by adding Gaussian noise in the normal bundle. We first construct an explicit coupling (Daskalakis et al., 2011; Roch, 2023) between $p_0 \mathcal{H}^m$ and $\tilde{p}_\sigma \mathcal{H}^m$.

Let $Z$ be a $\mathcal{M}$-supported random variable with law $p_0 \mathcal{H}^m$, i.e.

$$\mathbb{P}(Z \in A) = \int_A p_0(z) d\mathcal{H}^m(z), A \subset \mathcal{M}.$$

Conditionally on $Z = z$, draw Gaussian noise vector $N \in N_z \mathcal{M}$, $N \mid Z = z \sim \mathcal{N}(0, \sigma^2 I_{N_z \mathcal{M}})$, and set

$$X := Z + N \in \mathbb{R}^d, Y := \Pi(x) \in \mathcal{M}.$$

By construction, the law of $Z$ is exactly $p_0 \mathcal{H}^m$. Moreover, the law of $X$ is precisely the lifted distribution $p_\sigma(x)dx$: for any Borel set $B \subset \mathbb{R}^d$,

$$\mathbb{P}(X \in B) = \int_{\mathcal{M}} \mathbb{P}(Z + N \in B \mid Z = z) p_0(z) d\mathcal{H}^m(z) = \int_B p_\sigma(x) dx.$$

Since $Y = \Pi(x)$, its law is the pushforward of $p_\sigma(x)dx$ under $\Pi$, i.e.

$$\mathbb{P}(Y \in A) = \mathbb{P}(X \in \Pi^{-1}(A))$$
$$= \int_{\Pi^{-1}(A)} p_\sigma(x) dx$$
$$= (\Pi_\# p_\sigma)(A)$$
$$= \int_A \tilde{p}_\sigma(y) \mathcal{H}^m(y),$$

so $Y$ has law $\tilde{p}_\sigma \mathcal{H}^m$,

Therefore, the joint law of $(Z, Y)$, is a coupling of the probability measures $p_0 \mathcal{H}^m$ and $\tilde{p}_\sigma \mathcal{H}^m$ on $\mathcal{M}$.

It is standard (Daskalakis et al., 2011; Roch, 2023) that for any coupling $(U, V)$ of $\mu$ and $\nu$,

$$\|\mu - \nu\|_{\text{TV}} \leq \mathbb{P}(U \neq V).$$

In our setting,

$$\text{TV}(\tilde{p}_\sigma, p_0) = \|\tilde{p}_\sigma \mathcal{H}^m - p_0 \mathcal{H}^m\|_{\text{TV}} \leq \mathbb{P}(Z \neq Y).$$

We now bound $\mathbb{P}(Z \neq Y)$ using the reach of $\mathcal{M}$. Fix any radius $0 < r < \text{reach}(\mathcal{M})$ and recall that for $\mathcal{C}^2$ embedded manifolds, an equivalent characterization of the reach is that for all $z \in \mathcal{M}$ and all $n \in N_z\mathcal{M}$ with $\|n\| < r$, the point $x = z + n$ has $z$ as its unique nearest point on $\mathcal{M}$, hence $\Pi(z + n) = z$. In particular, if $\|N\| < r$ then $X = Z + N$ lies in the reach tube and

$$Y = \Pi(X) = \Pi(Z + N) = Z.$$

Therefore,

$$\{Z \neq Y\} \subseteq \{\|N\| \geq r\},$$

and hence

$$\mathbb{P}(Z \neq Y) \leq \mathbb{P}(\|N\| \geq r).$$

By construction, $N$ is a $k$-dimensional Gaussian in the normal space with law $\mathcal{N}(0, \sigma^2 I_k)$. The event $\{\|N\| \geq r\}$ is exactly the event that $X$ lies outside the tubular neighborhood of radius $r$. In particular,

$$\mathbb{P}(\|N\| \geq r) = \int_{\mathbb{R}^d \setminus T_r(\mathcal{M})} p_\sigma(x)dx.$$

Applying Lemma B.3 gives

$$\mathbb{P}(\|N\| \geq r) \leq C_1 e^{-C_2 \frac{r^2}{\sigma^2}}, \tag{3}$$

with $C_1 = 2^{\frac{(d-m)}{2}}$ and $C_2 = \frac{1}{4}$.

Combining the above inequalities gives

$$\text{TV}_{\mathcal{M}}(\tilde{p}_\sigma, p_0) \leq C_1 e^{-C_2 \frac{r^2}{\sigma^2}}.$$

$\square$

*Proof of Corollary 4.8.* By definition of the pointwise reach, for any $n \in N_z\mathcal{M}$ with $\|n\| < \tau(z)$, we have $\Pi(z + n) = z$. Remark that the pointwise reach tune is larger than or equal to the global reach tube. Hence,

$$\text{dist}(z, \Pi(z + n)) \mathbb{1}_{\{\|N\| < \tau(z)\}} = 0.$$

Therefore, we may define an expected pointwise displacement $D_\sigma(z)$ as

$$D_\sigma(z) = \mathbb{E}\left[\text{dist}(z, \Pi(z + n))\right] \leq 2\mathbb{E}\left[\|N\| \mathbb{1}_{\{\|N\| \geq \tau(z)\}}\right], \tag{4}$$

where the inequality follows from the triangle inequality and $z$ being a candidate point on $\mathcal{M}$:

$$\text{dist}\left(z, \Pi(z+n)\right) = \|\Pi(z) - \Pi(z+N)\| \leq \|z - (z+N)\| + \|(z+N) - \Pi(z+N)\| \leq 2\|N\|.$$

To bound the tail expectation, we again use the Gaussian tail estimate from Lemma B.3. We have $\mathbb{E}\|N\|^2 = \sigma^2 k$, and by Cauchy-Schwarz,

$$\mathbb{E}\left[\|N\|\mathbb{1}_{\{\|N\|\geq\tau(z)\}}\right] \leq \left(\mathbb{E}\|N\|^2\right)^{\frac{1}{2}} \mathbb{P}(\|N\| \geq \tau(z))^{\frac{1}{2}} \leq \sigma\sqrt{k}\mathbb{P}(\|N\| \geq \tau(z))^{\frac{1}{2}}. \tag{5}$$

By Equation (3), applied with $r = \tau(z)$, Lemma B.3 gives

$$\mathbb{P}(\|N\| \geq \tau(z)) \leq C_1 e^{-C_2 \frac{\tau(z)^2}{\sigma^2}}.$$

Substituting this into Equation (5), we obtain

$$\mathbb{E}\left[\|N\|\mathbb{1}_{\{\|N\|\geq\tau(z)\}}\right] \leq \sigma\sqrt{k}\left[C_1 e^{-C_2 \frac{\tau(z)^2}{\sigma^2}}\right]^{\frac{1}{2}}.$$

Combining this with (4) yields the pointwise bound

$$D_\sigma(z) \leq 2\sigma\sqrt{k}\left[C_1 e^{-C_2 \frac{\tau(z)^2}{\sigma^2}}\right]^{\frac{1}{2}}.$$

To obtain the integrated inequality, note that the nearest-point projection is $L$-Lipschitz on the pointwise reach tube, and therefore a pointwise displacement $D_\sigma$ induces an $LD_\sigma(z)$ deviation in densities after pushforward along $\Pi$. Integrating over $\mathcal{M}$ gives

$$\int_{\mathcal{M}} |\tilde{p}_\sigma(y) - p_0(y)| \, d\mathcal{H}^m(y) \leq 2L\sigma C_1 \int_{\mathcal{M}} e^{-C_2 \frac{\tau(z)^2}{\sigma^2}} \, d\mathcal{H}^m(z).$$

$\square$

# C. Experimental Setup

In this section, we provide more rigorous experimental details.

For all tasks, we use an Adam optimizer with a learning rate of $10^{-3}$. Gradients are clipped at maximum norms of 1.0, as is common in diffusion model training. We also uniformly set $T = 250$ across all diffusion model approaches. All experiments are done on NVIDIA A100 GPUs. We use a simple linear noise schedule with variances increasing from 0.0001 to 0.02, as in (Ho et al., 2020). For all tasks, a fixed number of samples are generated, and invalid samples such as those with NaN/Inf values are discarded before evaluation metrics are computed. In general, discarded samples make up usually no more than 10 out of $\mathcal{O}(10^4)$ samples, minimally affecting the reported results.

For PDM, the paper introducing the approach uses projections of scores. We project scores as approximated through the learned $\varepsilon$ outputs. For PIDM, we estimate intermediate samples during training as $\hat{x}_0 = \mathbb{E}\left[x_0 | x_t\right]$ using standard DDPM sampling.

We encountered several challenges and roadblocks implementing and accurately reproducing the results in RSGMs (Bortoli et al., 2022) due to out-of-date package versions and irreproducible environments. Reproducibility of the RSGM code is a documented issue. A concerted, good-faith effort was made to accurately implement the RSGM code, but ultimately yielded poor results with incompatibilities that would have been a perhaps unfair comparison. In any case, one can infer from the RSGM paper how many nontrivial computations are involved in both training and sampling. This may certainly be desirable for mathematical reasons, but can be non-trivially computationally prohibitive, as is noted by the RSGM authors themselves: a large part of the RSGM paper is focused on efficient techniques for reducing the computational cost of the many manifold-aware steps involved during both training and sampling. Due to these known reproducibility challenges and the unfairness of a comparison against a tailored intrinsic technique with high computational needs, we elect not to include it, or Riemannian Continuous Normalizing Flows (Mathieu & Nickel, 2020), as one of the comparison techniques in this work.

For the time embedding, we use concatenation of input samples with a predetermined time embedding. We use a simple concatenation of the scalar time step for the plane, sphere, and mesh tasks. The time embeddings for the image generation and protein tasks are formed through multi-layer perceptron (MLP) embeddings.

For each task, unless otherwise noted, the model parameterization (diffusion model or NF) and training-related hyperparameters are kept constant across all techniques to ensure a fair comparison. All NF approaches are run with learning rates of $10^{-4}$, batch sizes of 64, and a total number of 40 epochs during training. We use a RealNVP architecture with six affine coupling layers of hidden dimension 64 and a Glow architecture with hidden dimension 64 applied to interpret 3-D vector data as $1 \times 1$ images with three channels. For better insight into the neural network parameterized models used in each approach, we also report total parameter counts for each technique.

For the generation of uncertainty bands in Figures 2 and 6, we use 10 sampling trials per $\sigma$ for all experiments. In Figure 7, we use 3 trials per $(\sigma, \#\text{training samples})$ combination.

We note that the intrinsic metrics for the sphere task in Table 1 are computed with a Lambert azimuthal equal-area projection centered at the unit normal pole facing the top of the sphere.

For the mesh task, outliers with norm greater than 2.0 (only present for PIDM and DDPM) were filtered out for histogrammed metrics and visualization. The Gaussian is approximated through heat kernel sampling from a mean node along the geodesics induced by the edges of the shape mesh. The mesh (and thus points on the mesh) are centered at the origin and normalized in an aspect-ratio preserving way such that the maximum distance of a mesh point from the origin is 1.0.

|                   | **Plane** | **Sphere** | **Mesh** | **Images** | **Protein** |
|-------------------|-----------|------------|----------|------------|-------------|
| Time embedding    | Scalar    | Scalar     | Scalar   | MLP        | MLP         |
| Time emb. dim.    | 1         | 1          | 1        | 64         | 16          |
| # Training samples | 100,000  | 100,000    | 100,000  | 10,000     | 20,000      |
| Batch size        | 64        | 64         | 64       | 32         | 32          |
| Epochs            | 200       | 200        | 200      | 1000       | 1000        |
| Hidden dimension  | 64        | 64         | 128      | 1024       | 1024        |

*Table 6.* Combined diffusion model training configurations for all tasks.

| # Parameters | Plane | Sphere | Mesh | Images | Protein |
|---|---|---|---|---|---|
| RealNVP | 42,780 | 42,780 | – | – | – |
| Glow | 28,836 | 28,836 | – | – | – |
| DDPM | 4,675 | 4,675 | 17,539 | 1,093,057 | 112,353 |

*Table 7.* Total number of trainable parameters for each model across all tasks.

# D. Evaluation Metrics

## D.1. Point Cloud Evaluation Metrics

We present definitions of all metrics used in Section 5.

**Definition D.1** (Coverage (COV)). Let $\{x_i\}_{i=1}^n$ denote the set of real samples and $\{y_j^n\}_{j=1}^m$ denote the set of generated samples. For each $x_i$, let $r_i$ be the distance to its $k$-th nearest neighbor among real samples $\{x_l\}_{l=1}^n$ and compute the distance to the nearest $y_j$:

$$d_i = \min_{1 \leq j \leq m} \|x_i - y_j\|_2.$$

We say that a real sample $x_j$ is *covered* if $d_i \leq r_i$. Then, we define the coverage metric as the fraction of real samples that are covered:

$$\text{COV}(X, Y) = \frac{1}{n} \sum_{i=1}^n \mathbb{1}_{d_i \leq r_i}.$$

For the empirical Jensen-Shannon Divergence (JSD) and Total Variation Distance (TVD) metrics, we bin samples into histograms, where the number of bins in each axis is fixed for each approach. For the plane and sphere tasks, we use 25 bins in each direction. For the mesh task, we use 50 bins. In all cases, $p_0$ is used as the reference distribution.

## D.2. Image Evaluation Metrics

Successfully evaluating distributions of structured images in such a way that avoids the pitfalls of high dimensionality and respects invariances is a known challenge (Jayasumana et al., 2024; Benny et al., 2021). To combat this, many works in image generation employ embedding-based distances, wherein a secondary classification model is trained (or a benchmark model is reloaded) and latent embeddings are extracted from intermediate layers to be used as lower-dimensional embeddings of the input data samples. The Frechét Inception Distance (FID) score (Heusel et al., 2017) is one example of this, where distances between image distributions are evaluated by studying statistics of the embeddings from the popular Inception-V3 architecture. The idea is that a well-trained model will have learned embeddings which capture image invariances and essential structure, enabling the statistics of such embeddings to act as useful proxies for the data distribution.

For the MNIST benchmark dataset, the issue of image classification is widely considered to be a solved problem in modern machine learning. To that end, we train a straightforward LeNet-style classifier (Lecun et al., 1998) with 50,190 learnable parameters using a 90%/10% training/validation split over the 60K MNIST training dataset images. The penultimate embeddings are used to compute an FID-like embedding score as follows:

**Definition D.2.** Define $f_\varphi : \mathbb{R}^{28 \times 28}$ as the embedding map, where we let $d = 84$ and define the embedding as the latent vector of the neural network classifier at the penultimate layer. Let $\{x_i\}_{i=1}^N \sim P$ denote true samples and $\{y_j\}_{j=1}^M \sim Q$ denote generated samples. Compute their embeddings $z_i := f_\varphi(x_i)$ and $w_j := f_\varphi(y_j)$ and obtain empirical means and covariances of each collection of embedded samples as $(\hat{\mu}_P, \hat{\Sigma}_P)$ for the true samples and $(\hat{\mu}_Q, \hat{\Sigma}_Q)$ for the generated samples.

From these empirical means and covariances, we compute an FID-like score by computing the Fréchet distance between the empirical Gaussians:

$$\text{FID}_f(P, Q) := \|\hat{\mu}_P - \hat{\mu}_Q\|_2^2 + \text{Tr}\left(\hat{\Sigma}_P - \hat{\Sigma}_Q - 2\left(\hat{\Sigma}_P\hat{\Sigma}_Q\right)^{\frac{1}{2}}\right). \tag{6}$$

The metrics we report in Section 5.4 are using the $\text{FID}_f$ score between the heldout MNIST test set and 1,000 generated samples.

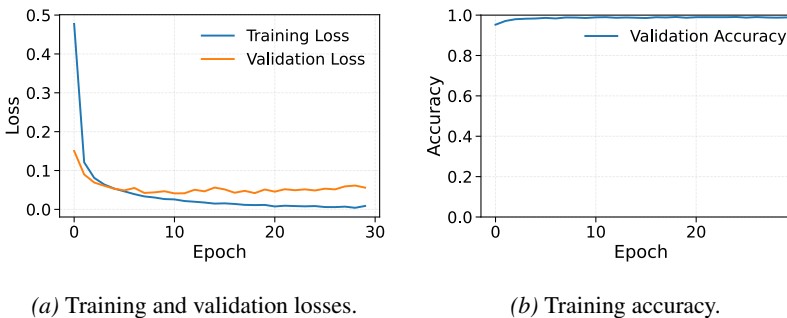

*(a)* Training and validation losses.   *(b)* Training accuracy.

*Figure 9.* Training and validation losses and accuracy for MNIST classifier.

An advantage of the image generation task under fixed total flux case is its easy visual interpretability. We present several samples from each generated image distribution in Appendix G to be analyzed in conjunction with the learned embedding metrics.

The standard MNIST dataset also provides balanced classes, with roughly 10% of all data samples allocated to each digit class. Therefore, one would naturally expect that a well-trained generative model would also have similar class distribution. Again, we utilize the above trained classification model to identify class labels for all generated samples from each generated distribution, and interpret the class distributions of the generated samples against those of the underlying distribution as a measure of distributional accuracy.

**Definition D.3** (Class JSD). Define the trained MNIST classifier as $g : \mathbb{R}^{784} \to \{0,1,2,3,4,5,6,7,8,9\}$. Compute predicted class labels for a set of generated samples $\{x_i\}_{i=1}^N$ as $\hat{y}_i = g(x_i)$. This forms an empirical class distribution $\hat{p}_k^{\text{emp. class}} := \frac{1}{N} \sum_{i=1}^N \mathbb{1}_{\{\hat{y}_i = k\}}$ for $k \in \{0,1,2,3,4,5,6,7,8,9\}$. Let the class distribution of the data be $p^{\text{data class}}$. Then, we report the *class JSD* as the JSD between $\hat{p}^{\text{emp. class}}$ and $p^{\text{data class}}$.

We note, however, that this evaluation metric is inherently hindered by the classification model's inability to determine out-of-distribution samples, as illegible generated samples will still receive a class label. We again recommend the reader to look at these metrics in conjunction with the image examples in Appendix G.

### D.3. Protein Evaluation Metrics

Each protein sample has dimensionality $x \in \mathbb{R}^{90}$. As in the image case, this dimensionality hinders the effective usage of histogramming-based evaluation metrics. We present alternative metrics intended to measure the diversity

**Definition D.4** (Pairwise RMSD). Denote $\{x_i\}_i^n$ as a set of generated protein backbone samples, for which each sample $x_i = \text{vec}(X_i)$ where $X_i \in \mathbb{R}^{3N}$ contains the coordinates of $N$ atoms, and pick any pair $(i,j)$ with $i \neq j$. Let the root-mean-square deviation (RMSD) between samples $x_i$ and $x_j$ after optimal rigid alignment be

$$\text{RMSD}(x_i, x_j) := \min_{R \in \text{SO}(3), K \in \mathbb{R}^3} \left( \frac{1}{N} \sum_{n=1}^N \left\| X_i^n - (RX_j^{(n)} + K) \right\|_2^2 \right), \tag{7}$$

where $R$ and $K$ are a rotation and a translation, respectively, which are obtained using the Kabsch algorithm. We define the *pairwise RMSD distribution* of the set of generated samples as the collection of all RMSD values over all possible unordered pairs. For the pairwise RMSD metrics presented in Section 5.5, we report the median over the pairwise RMSD distribution. Higher values suggest greater diversity of structures within the collection of generated samples. Lower values suggest possible mode collapse.

**Definition D.5** (Maximum Mean Discrepancy). We use the Maximum Mean Discrepancy (MMD) metric, which compares distributions using kernel mean embeddings in a Reproducing Kernel Hilbert Space (RKHS). We present the squared MMD with a Radial Basis Function (RBF) kernel:

$$\hat{\text{MMD}} = \frac{1}{N^2} \sum_{i,i'} k(x_i, x_{i'}) + \frac{1}{M^2} \sum_{j,j'} k(y_j, y_{j'}) - \frac{2}{NM} \sum_{i,j} k(x_i, y_j).$$

In our results, we use a bandwidth of 1.0.

# E. Sampling Stability

In this section, we present the same results as in Figure 4 for the remaining tasks (mesh, image, and protein backbone). This empirically demonstrates the known issue of score explosion for diffusion model sampling on manifolds (Bortoli, 2022) and log-determinant explosion for normalizing flows and demonstrates how our proposed approach combats these issues. We remark that samples leading to NaN outputs were removed from tracking. However, they are included in the following section regarding the number of observed NaN/Inf samples associated with each method.

In particular, we frequently observe a continuous reduction in the median magnitude of scores as $\sigma$ increases.[1] This is expected, as $\sigma$ increasing means that the nonzero region of $p_\sigma$ occupies an increasingly larger space in $\mathbb{R}^d$. We use median aggregation of scores instead of average aggregation of scores in order to avoid outliers biasing the results.

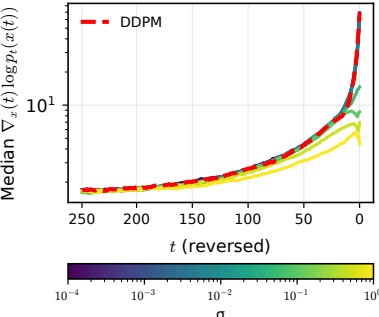

*Figure 10.* Mesh

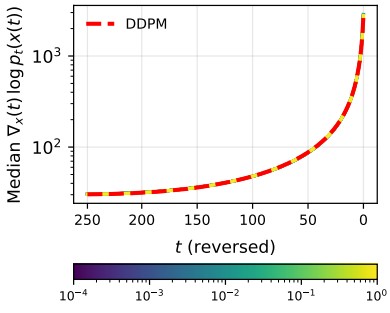

*Figure 11.* Image

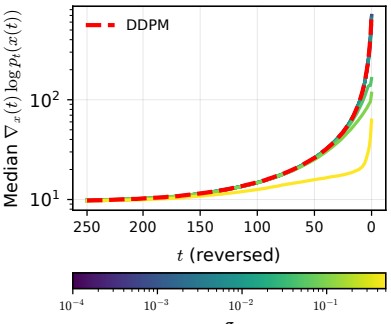

*Figure 12.* Protein

[1]We do not claim that larger scores are inherently "bad." However, larger scores run the risk of numerical instability or overflow on low-precision training or sampling, the latter of which is growing increasingly common in generative AI applications. Therefore, obtaining competitive performance with bounded scores (as is offered by our proposed $p_\sigma$ approach) is highly desirable.

# F. Manifold Implementations

## F.1. Plane Constraint

For the plane task, the manifold is simply an affine subspace

$$\mathcal{M}_{\text{plane}} = \{x \in \mathbb{R}^3 : Ax = b\},$$

where $A \in \mathbb{R}^{1 \times 3}$ is a row vector normal to the plane and $b \in \mathbb{R}$ is an offset. The nearest–point projection map $\Pi(x)$ has a closed form:

---
**Algorithm 3** Nearest-Point Projection onto $\mathcal{M}_{\text{plane}}$

---
**Require:** Query point $x \in \mathbb{R}^3$, row vector $A$, offset $b$
  1: Compute projection matrix $P = A^\top (AA^\top)^{-1} A$
  2: Compute offset $o = A^\top (AA^\top)^{-1} b$
  3: **Return:** $\Pi(x) = (I - P)x + o$

---

This expression is exact and requires only a pseudoinverse of $AA^\top$.

## F.2. Sphere Constraint

For the sphere task, the manifold is a quadratic surface

$$\mathcal{M}_{\text{sphere}} = \{x \in \mathbb{R}^3 : \|x - c\|_2 = r\},$$

where $c \in \mathbb{R}^3$ is the center and $r > 0$ the radius. The projection map $\Pi(x)$ rescales the radial direction:

---
**Algorithm 4** Nearest-Point Projection onto $\mathcal{M}_{\text{plane}}$

---
**Require:** Query point $x \in \mathbb{R}^3$, sphere center $c$, radius $r$
  1: Compute direction $d = x - c$
  2: Normalize direction $u = \frac{d}{\|d\|_2}$
  3: **Return:** $\Pi(x) = c + ru$

---

This projection is also exact, as each point is simply snapped radially onto the sphere surface.

## F.3. Mesh

Define the manifold as

$$\mathcal{M}_{\text{mesh}} = \{x \in \mathbb{R}^3 : h(x) = 0\},$$

where $h(x)$ is the level set defining the surface of the mesh. We implement an approximate nearest-point projection map $\Pi(x)$:

---

**Algorithm 5** Nearest-Point Projection onto $\mathcal{M}_{\text{mesh}}$

---

**Require:** Query point $x \in \mathbb{R}^3$, mesh $\mathcal{T}$ with vertices $V$ and faces $F$
1: Compute $k$ nearest vertices $v_1, ..., v_k \in V$
2: Gather incident faces for each $v_i$
3: Rank candidate faces by distance from $x$ to their centroids
4: keep closest $m$ faces
5: **for** each candidate triangle $\tau_j$ with vertices $(a, b, c)$ **do**
6:     Compute closest point $y_j$ on $\tau_j$ using an exact point-triangle projection
7:     Record $d_j^2 = \|x - y_j\|^2$
8: **end for**
9: Let $j^* = \arg \min_j d_j^2$
10: **Return:** $\Pi(x) = y_{j^*}$

---

This procedure guarantees that $\Pi(x)$ lies on $\mathcal{M}_{\text{mesh}}$. The parameters $k$ and $m$ trade off accuracy and cost: larger values increase the candidate pool size and robustness, while smaller values reduce the computational needs. In all of our experiments, we use $k = 2$ and $m = 16$.

### F.4. Images

We construct a variant of the MNIST dataset constrained by total image flux by embedding each image into the manifold

$$\mathcal{M}_{\text{flux}} = \{x \in \mathbb{R}^{784} : \mathbf{1}^\top x = s\},$$

where $\mathbf{1} \in \mathbb{R}^{784}$ is a vector of all ones and $s \in \mathbb{R}$ is the total pixel sum. The constraint Jacobian may be written as

$$J_g(x) = \nabla h(x)^\top = \mathbf{1}^\top \in \mathbb{R}^{1 \times 784}, h(x) = \mathbf{1}^\top x - s,$$

constructing tangent and normal spaces

$$T_x\mathcal{M} = \{v \in \mathbb{R}^{784} : \mathbf{1}^\top v = 0\}, N_x\mathcal{M} = \text{span}\{\mathbf{1}\},$$

with the unit normal vector $\hat{n} = \frac{1}{\sqrt{784}} = \frac{1}{28}$. The orthogonal projector onto $T_x\mathcal{M}$ may be written as

$$P_T = I - \frac{1}{784}\mathbf{1}\mathbf{1}^\top$$

and the orthogonal projector onto $N_x\mathcal{M}$ is

$$P_N = \frac{1}{784}\mathbf{1}\mathbf{1}^\top.$$

Samples from $p_\sigma$ may be formed by perturbing each sample from $p_0$ with Gaussian noise along the normal direction:

$$x_\sigma^i = x_0^i + \frac{\varepsilon}{28}\mathbf{1}.$$

We may write the nearest-point projection onto $\mathcal{M}_{\text{flux}}$ as

$$\Pi(x) = \arg \min_{y \in \mathcal{M}} \frac{1}{2}\|y - x\|^2 = x + \frac{s - \sum_{i=1}^{784} x[i]}{784}\mathbf{1} = \left(I - \frac{1}{784}\mathbf{1}\mathbf{1}^\top\right) x + \frac{s}{784}\mathbf{1}.$$

### F.5. Protein Generation Problem

Let $x_{i,a} \in \mathbb{R}^3$ denote the coordinates of atom $a \in \{\mathrm{N}, \mathrm{CA}, \mathrm{C}\}$ in residue $i$.

We impose two types of local geometric constraints for the purposes of this task: bond length constraints and bond angle constraints.

Let $\mathcal{E}_{\mathrm{len}}$ and $\mathcal{E}_{\mathrm{ang}}$ denote the index sets for bond length (pairs of atoms) and bond angle (triplets of atoms) constraints, respectively. For each $e = ((i,a),(j,b)) \in \mathcal{E}_{\mathrm{len}}$ construct the bond length residual as

$$r_e(x) = \|x_{i,a} - x_{j,b}\|_2 - \ell_{ab},$$

where $\ell_{ab}$ is the bond length constraint for the atoms indexed by $a$ and $b$. For each $e = ((i,a),(j,b),(k,c)) \in \mathcal{E}_{\mathrm{ang}}$, construct the bond angle residual as

$$r_e(x) = \cos\theta_e(x) - \cos\theta_e^*, \cos\theta_e(x) = \frac{(x_{i,a} - x_{j,b}) \cdot (x_{k,c} - x_{j,b})}{\|x_{i,a} - x_{j,b}\|_2 \|x_{k,c} - x_{j,b}\|_2}.$$

Collecting all such bond length and bond angle residuals gives

$$c(x) = (r_e(x))_{e \in \mathcal{E}_{\mathrm{len}} \cup \mathcal{E}_{\mathrm{ang}}}.$$

The protein backbone constraint manifold is thus the zero level set

$$\mathcal{M}_{\mathrm{protein}} = \{x \in \mathbb{R}^D : c(x) = 0\}.$$

To project samples to $\mathcal{M}_{\mathrm{protein}}$, we use a Gauss-Newton optimization algorithm with a maximum number of 5 iterations, a step size of 1.0, a damping factor of $10^{-4}$, and a tolerance maximum residual of $10^{-4}$.

# G. Images with Total Flux Samples

We present additional sample images from each generated distribution at various training data sample set sizes and $\sigma$ to provide improved qualitative insight into the sample quality from each learned distribution, complementing the quantitative metrics presented in Section 5.4.

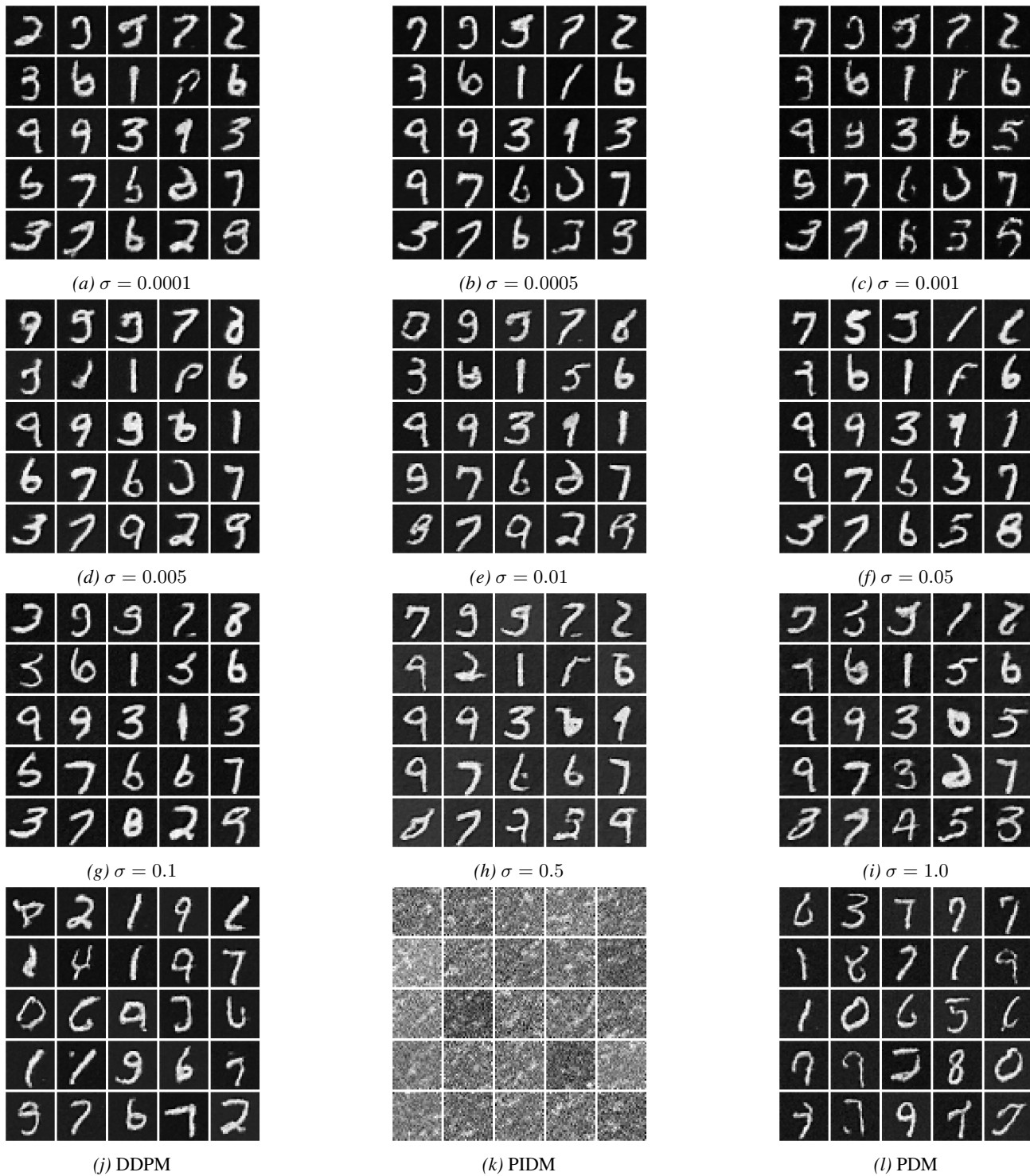

*Figure 13.* Examples of effect of $\sigma$ with 100 training data samples.

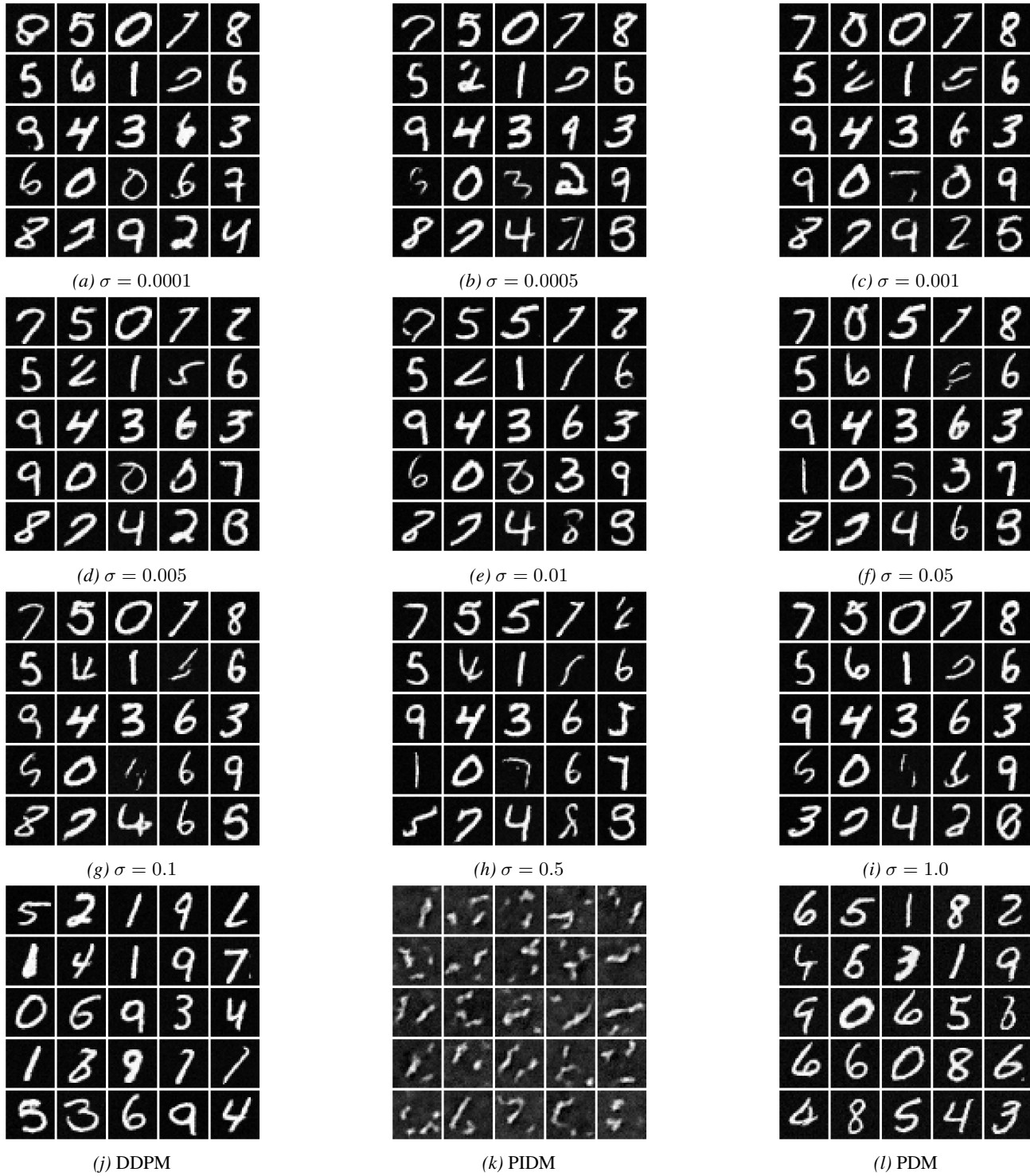

*(a) σ = 0.0001*

*(b) σ = 0.0005*

*(c) σ = 0.001*

*(d) σ = 0.005*

*(e) σ = 0.01*

*(f) σ = 0.05*

*(g) σ = 0.1*

*(h) σ = 0.5*

*(i) σ = 1.0*

*(j) DDPM*

*(k) PIDM*

*(l) PDM*

*Figure 14.* Examples of effect of σ with 1,000 training data samples.

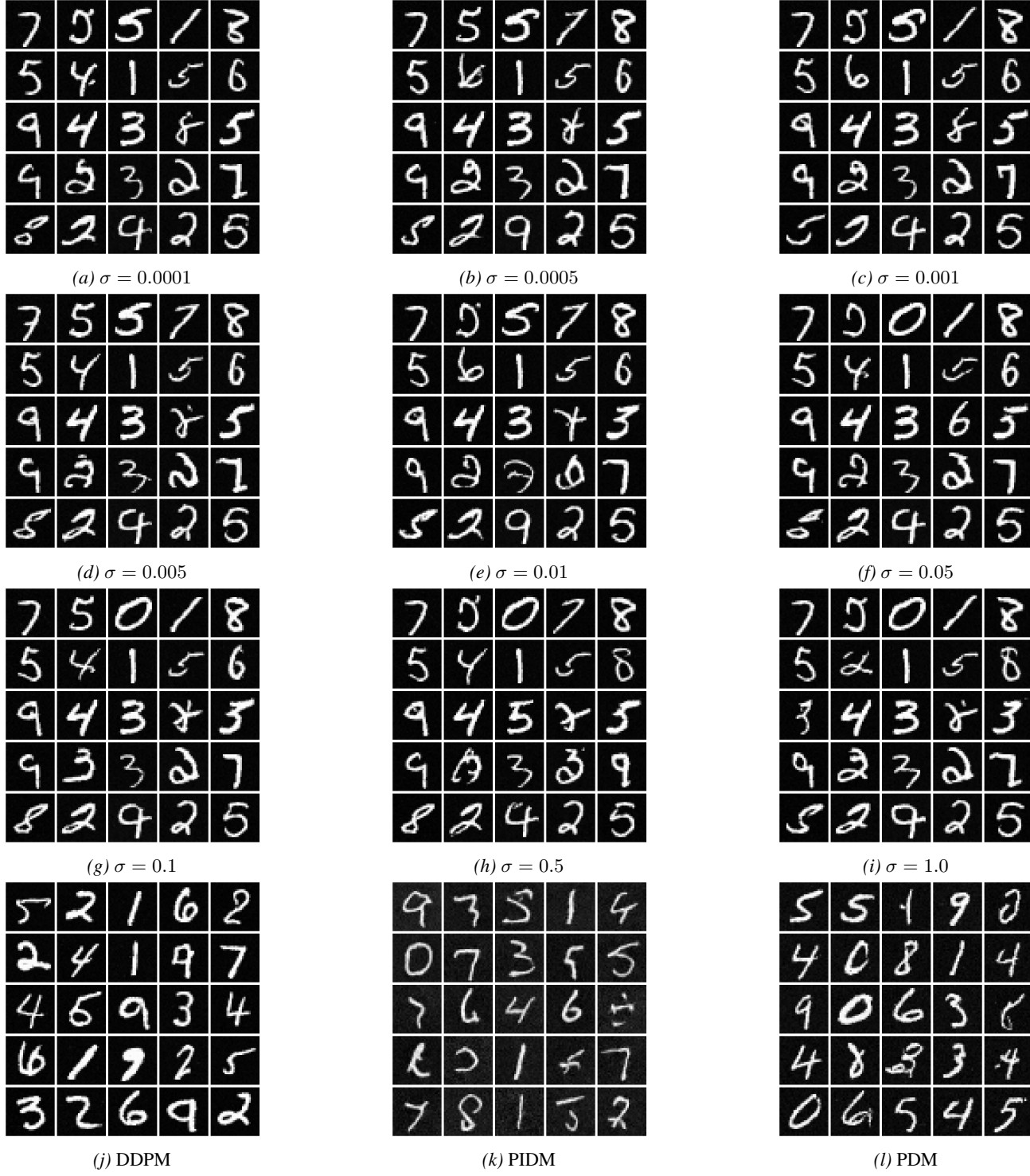

Figure 15. Examples of effect of $\sigma$ with 10,000 training data samples.

