# OpenReview forum: "Manifold-Aware Perturbations for Constrained Generative Modeling"
_ICML.cc/2026/Conference — ICML 2026 spotlight_

### Official Review · Reviewer_4nzA · 2026-03-08

**Soundness:** 3
**Presentation:** 4
**Significance:** 4
**Originality:** 4
**Overall Recommendation:** 5
**Confidence:** 2

**Summary:**

This work introduces a new method to modify the underlying data-distribution learned by a generative model addressing common limitations in generative models for equality-constrained distributions. It proposes perturbing data strictly in the normal direction of the manifold to preserve geometry while matching ambient space dimensions,  enabling stable training and sampling with diffusion models and normalizing flows. A theoretical analysis providing guarantees like perfect recovery for linear constraints,  total variation error bound for nonlinear constraints, bounded scores, and non-zero Jacobian determinants is provided together with extensive empirical validation on a good number of representative tasks to support the proposed method.

**Compliance With Llm Reviewing Policy:**

Affirmed.

**Final Justification:**

My final recommendation is 5: Accept. The presentation is excellent and accessible for both theoretical and experimental audiences.  The authors successfully address two important and common challenges appearing in prior works: how to limit distortion of the original data distribution and how to reduce the error in the induced distribution after projection. The two remarks in answer 1 of the rebuttal were very insightful. The rebuttal reinforced my prior assessment.

**Key Questions For Authors:**

(1) Theorem 4.6: When the reach of the manifold is small,  the exponential decay in the bound gets worse and small projection errors can intensify the distortion. If we put $\sigma$ very small following your recommendation (line 179), then we don't get the benefits your proposed method anymore. How can your proposed method mitigate this situation?

(2) In the case of Score-based Diffusion models, which (theoretical) assumptions need to be imposed on the score network if we use your perturbation? Is the score network more difficult to choose for "approximating" $\nabla \log p_{\sigma}$?

(3)  I would suggest stating the assumptions on the manifold in lines 141-143 (page 3), line 155 ($p_0$ Lipschitz on the manifold) and lines 156-157  in a separate assumption environment and report them in the statement of Theorem 4.5/4.6 and Corollary 4.7.

**Limitations:**

Yes

**Strengths And Weaknesses:**

Strengths:

(1) This work relies on a novel way to construct anisotropic noising of the original data distribution in careful way such that it keeps the distortion of the original data distribution limited and reduces the error in the induced distribution after projection, mitigating these common two important shortcomings appearing in prior works.

(2) It provides theoretical guarantees of the original data distribution recovery from the proposed modification of the data distribution.

(3) The manuscript is well-written and struck a good balance in terms of readability for a mathematically-sounded audience (error analysis section) and more experimental-prone audience.

Weaknesses:

(1) The proposed method suffers when the reach of the manifold is small due to high curvature. See questions to authors.

(2) A discussion about how the proposed modification of the data distribution may change the training of the generative model (Score-based diffusion model and Normalizing Flow) is missing. See questions to Authors.

---

> ### Author Rebuttal · Authors · 2026-03-31
>
> We thank the reviewer for their perspectives! We address the key questions below.
>
> 1 - We make two remarks: we only need $\sigma$ to be small relative to the reach of the manifold. With access to a projector (generally feasible for known constraint manifolds and an appropriate optimization algorithm), our bound holds for manifolds with small reach, although this is certainly the case where the bound becomes looser. For small projection errors possibly inducing further distributional error under small reach, we note that this is only problematic if nontrivial probability mass is present in the regions of small (local) reach. Challenges where many projection errors may be possible are certainly present in science and engineering, but we conjecture that for the well-behaved manifolds we consider within the scope of this paper, existing state-of-the-art optimization techniques which admit nearest-point projection algorithms are sufficient to successfully return samples to the constraint manifold without overly amplifying distributional errors.
>
> 2 - No additional theoretical assumptions need to be imposed on the score network under our proposed perturbation. In fact, theoretically, the score network should be able to approximate the score more easily under our proposed distributional modification. For manifold-constrained distributions, the score field should go to infinity as we spatially approach the manifold at small t, which can lead to high numerical instabilities and is generally known to be challenging for the score network to approximate. Under our proposed modification, the score network should not have to learn to approximate infinity under the standard training objective, so the usual choices for selecting a score network architecture or regularization can be made and no additional assumptions need be imposed.
>
> 3 - We have made the suggested change for clarity in the theory section. Thank you for the suggestion!

---

> > ### Author Rebuttal · Reviewer_4nzA · 2026-04-01
> >
> > Thank you for your replies.
> >
> > I  have no further questions. I am satisfied with the authors' responses. I therefore maintain my Overall Recommendation: 5: Accept unchanged.

---

### Official Review · Reviewer_FbC2 · 2026-03-09

**Soundness:** 2
**Presentation:** 3
**Significance:** 2
**Originality:** 2
**Overall Recommendation:** 4
**Confidence:** 4

**Summary:**

The paper studies the problem of training generative models when the data distribution is supported on a lower-dimensional manifold embedded in a higher-dimensional ambient space. In such cases, the true data distribution is singular with respect to the Lebesgue measure, which creates difficulties for bijective generative models such as normalizing flows and diffusion models.

To address this issue, the authors propose to perturb the data before training by adding Gaussian noise restricted to the normal space of the manifold. The model is trained on this perturbed distribution, and generated samples are subsequently projected back onto the manifold to recover the original data distribution. The approach assumes that the data manifold and its normal space are known a priori.

The paper provides a theoretical analysis of the distortion between the recovered distribution and the true data distribution. In particular, the authors derive bounds on the total variation distance between the two distributions, showing that the distortion depends on the reach of the manifold and the noise level. The analysis implies that the distortion vanishes for linear manifolds and remains small when the noise standard deviation is significantly less than the reach of the manifold.

The proposed approach is evaluated using normalizing flows and diffusion models on several synthetic and real-world datasets where the data lies on known manifolds, including planar and spherical manifolds, an image generation task with a flux constraint, and a protein backbone generation task. The results suggest that learning the perturbed distribution and projecting samples back onto the manifold can improve the recovery of the target distribution compared to alternative approaches.

**Compliance With Llm Reviewing Policy:**

Affirmed.

**Final Justification:**

The authors proposed considerable improvements during the rebuttal, so I increased my score to "4: weak accept". However, the experiments are still somewhat weak and could be much more convincing.

**Key Questions For Authors:**

see above.

**Limitations:**

The practical relevance of the idea, including its limitations, should be discussed more in depth, especially by using more convincing experiments.

**Strengths And Weaknesses:**

Strengths:
* The paper addresses a well-known limitation of (likelihood-based) generative models when the data distribution is supported on a lower-dimensional manifold. The problem is introduced clearly and the motivation is well explained.
* The authors suggest to fix this problem by perturbing the data with noise only in the normal space rather than the more common isotropic noise. The text clearly motivates why this is beneficial and provides theoretical analysis and empirical evidence to support this claim. The total variation bound relating the TV error of the learned distribution to the ratio between the manifold's reach and the noise standard deviation is the central result.
* The discussion of the differences between densities defined with respect to the Lebesgue measure versus the Hausdorff measure is helpful and clarifies an important conceptual point when dealing with manifold-supported distributions.

Weaknesses:
* While the reach is an interesting characteristic of a manifold, it is also very pessimistic: global reach is only tight at the point of maximal curvature, and local reach is only tight in the direction of the nearest medial axis. At most locations and in most directions, the effective reach is much higher, especially when the embedding space is high-dimensional. On the other hand, it is known that the tangential component of isotropic noise is often negligible when the noise is small, so that strictly normal noise has no measurable benefit. It is therefore not clear if theorem 4.6 - the paper's central theoretical result - provides a relevant bound in practice. Historically, bounds from classical learning theory often turned out to be vacuous because real-world problems tend to be far from the assumed worst case. The relevance of the bound should be established experimentally, ideally with real-world examples rather than toy problems.
* If the noise has bounded norm instead of being Gaussian, one can adapt the noise level globally or locally to never exceed the global or local reach. This possibility should be discussed.
* Are situations where the manifold is known (so that one can restrict the noise to the normal space and perform an exact projection) common in practice? The datasets in the experimental section are rather simple (sphere, plane) or contrived (linear constraint on MNIST).  The only exception is the Protein Backbone Generation (Section 5.5), but the setup is insufficiently specified here. It remains unclear how the manifold is constructed and, more importantly, how the normal space $N_z \mathcal{M}$ required for the noise injection is computed (the additional explanations in appendix F.5 do not fully clarify this). More examples with clear real-world benefits would be much more convincing.
* If the manifold is not known, is there a way to approximate noise augmentation in the normal space with reasonable effort and accuracy?
* The paper misses important prior work that should be included in the experiments, and - where applicable - in theoretical comparisons., e.g.
   - Alain & Bengio (2014): What Regularized Auto-Encoders Learn from the Data-Generating Distribution (precise quantification of the effects of isotropic noise)
   - Rifai et al. (2011): Contractive auto-encoders: explicit invariance during feature extraction (explicitly penalize the tangent space)
   - Sorrensen et al. (2024): Learning distributions on manifolds with free-form flows (an alternative method to restricting the distribution to a known manifold)
   - Jacobson et al. (2025): Staying on the Manifold: Geometry-Aware Noise Injection (clear separation between tangential and normal noise components)
   - Kim et al. (2020): SoftFlow (is cited in related work, but immediately dismissed -- should instead be included in the experiments)
* Eq. (1) is not correct. As written, $x$ is the latent variable, and $f^{-1}(x)$ is the decoder. To be consistent with this notation, the determinant must be computed from the Jacobian of $f^{-1}$, not $f$, and it belongs to the LHS of the equation (when written with exponent $-1$). Moreover, when $p_0$ is restricted to a manifold (the case of interest here), the Jacobian is rectangular and has no determinant. Instead, the determinant term becomes $\sqrt{\det J^T J}$.
* Metrics such as COV and JSD are used without explicit definition in the main text. It is also unclear whether the reference distribution for these metrics is $p_0$ or the perturbed distribution $p_{\sigma}$​.
* Section 5.3 does not specify what the distribution to be learned is and how the errors are computed. Table 2 is never referenced in the text and not properly explained. The same holds for figure 4. The caption of figure 3 is ambiguous, for example "learning $p_\sigma$ consistently outperforms learning $p_0$ and post-projecting samples" - isn't post-projection of samples a key ingredient of the proposed method? The caption of figure 6 claims "learning $p_\sigma$ consistently improves upon the projected DDPM baseline on all advanced problems", when the baseline actually performs comparably in several experiments. The meaning of the “average Jacobian log-determinant” in Fig. 2 is unclear (e.g., whether it refers to the encoder or decoder Jacobian).
* l.218 right: If the label “$p_{\sigma}$” refers to projected samples, instead use the symbol $\tilde{p}_{\sigma}$ that had been introduced exactly for this purpose.
* The figures and tables mix results from different training protocols (e.g., isotropic noise vs. normal-space noise) without explicitly stating which variant is used for each method, making the comparison difficult to interpret.
* The paper states that naïve training of $p_0$​ leads to “exploding log-determinants” (l.19 right), which is presented as a key motivation​. The experiments do not explicitly demonstrate this behavior and quantify the consequences. Similarly, the claim of “improved Jacobian determinant stability” (l.213 right) is not clearly supported by the results shown in Fig. 2. A decrease/increase of the average log-determinants is not very informative and only indicates that the model is trained on data of smaller/higher entropy. Analyzing the variance or spatial stability of the log-determinants across samples or training runs would provide more insight.
* The concept of the reach tube is used in multiple places but is not clearly defined in the main text, and phrases such as “specific tubes around M” remain vague for readers unfamiliar with the topic.
* Some sections could be simplified or shortened.

---

> ### Author Rebuttal · Authors · 2026-03-31
>
> We thank the reviewer for their insights! We address weaknesses in enumerated order.
>
> 1 - It is true that the global reach is pessimistic, hence our local reach corollary. However, it is not true that strictly normal noise has no measurable benefit. We also note that our bound also depends on the codimension of M and not necessarily the data (intrinsic or extrinsic) dimensionality. In regards to the point about the role of $\sigma$, this is done in the $\sigma$ sweeps for each task, but we are working on aligning the experiment discussion with our theory.
> 2 - This is an interesting point! Alternative mean-zero bounded norm noising strategies could certainly be employed. We have included this as future work.
> 3 - We note that we refer to “manifold” and “constraint” somewhat interchangeably. Constraints are abundant in any case where aligning samples with domain knowledge is critical. We draw the analogy to cases which utilize PINNs: if constraints weren’t known, then residuals would not be able to be computed. Analogously, the use case of our paper is the setting where scientists may wish to use generative modeling for their applications where samples are known to be constrained. Thus, the case where M is known is highly common in practice, as evidenced by the abundance of research necessitating training on constraint residuals.
> We present highly realistic testbeds: the plane/sphere are illustrate the anticipated effects of our theory do hold for M of infinite/finite reach, although affine constraints are common in real-world problems and the plane already illustrates that our method only helps with no performance downsides in the linear case (to connect this to real problems, one may think of linear inverse problems as data constrained by affine functions). The mesh task is inspired by a task in Elhag et al. (2023) and serves to extend the plane/sphere results and show that they are robust under high and irregular curvature. The MNIST example is actually quite realistic and is derived from Feng et al. (2024)’s paper, “Neural Approximate Mirror Maps,” which, while taking a different approach to constrained generative modeling, experiments on total-flux-constrained images and cites the Event Horizon Telescope (EHT) collaboration as a direct application in scientific inverse imaging where flux is known to be constrained. The EHT is a highly relevant domain, so our experimental task of constrained imaging is quite pertinent.
> We are writing a substantially more fleshed out version of Appendix F.5 with clear descriptions of noising and projection for the protein task.
> 4 - We only consider the case where the constraints are known, so as to better align generative modeling strategies with domain knowledge. However, the approximate M case is possible. We are investigating this in follow-on work, but note that the 1st stage of our method only requires knowledge of M’s normal vectors, and not necessarily even M itself. Therefore, imprecise approximations of M or small offsets can be acceptable so long as we have a good sense of manifold curvature, at least up to 1st-order.
>
> 5 - We are investigating the first two papers. The third was studied during the writing of the paper but omitted as it would not be a fair comparison. The fourth is not a generative modeling strategy and is a technique for supervised learning. The fifth is implicitly included as our isotropic noising comparisons are intended as a catch-all for isotropic noising strategies, and SoftFlow does not offer any mathematical guarantees as ours does. We will modify our discussion of the related literature to make this clear.
> 6 - We note the confusing notation and have reviewed it in the paper.
> 7 - We state in the beginning of Section 5 that all metrics are defined in Appendix D for space reasons. $p_{0}$ is used as the reference distribution in all cases. We have edited Appendix D.
> 8 - We state that the distribution being learned is a Gaussian of 3-D points supported on M’s surface. We have clarified how the errors are computed for the mesh task in 5.3. Table 2 and Figure 4 are now referenced in the text. The caption of figure 3 is explaining that our method outperforms the naive baseline of learning $p_{0}$ and simply projecting samples back to the manifold. We are referring to the Jacobians of the NF models.
> 9 - Addressed.
> 10 - We are unsure what is meant by “which variant is used for each method” and ask for clarification. Thank you very much!
> 11 - We do observe high (not 1, which would indicate invertibility) log-determinants in the NF figures, which show the motivating phenomenon. We have rewritten the noted phrase as “decreased Jacobian determinant magnitude,” which is indeed observed. The model is indeed trained on data of changed entropy, which is done in a manifold-aware way by construction. The variance of the magnitude is indeed an interesting quantity which we plan to investigate.
> 12 - Addressed.
> 13 - We are refining the paper’s succinctness.

---

> > ### Author Rebuttal · Reviewer_FbC2 · 2026-04-02
> >
> > 1. You state in the rebuttal that your method is highly relevant in practice, because "Constraints are abundant in any case where aligning samples with domain knowledge is critical." However, while simple manifolds (e.g. plane, sphere) are available in closed-form, constraints typically define the manifold indirectly via a set of implicit equations. In order to demonstrate the relevance of your method for this very common situation, it would be great if you provided (and included in the paper) a general recipe and corresponding formula and algorithms to *add noise only in the normal directions, when the manifold is defined by a set of implicit equations.* I'll definitely raise my score then.
> >
> > 2. In the meantime, I noticed that the paper "Density estimation on low-dimensional manifolds: an inflation-deflation approach"  (https://www.jmlr.org/papers/v24/21-0235.html) is also highly relevant prior work. Please comment on it as well.
> >
> > 3. "We are unsure what is meant by 'which variant is used for each method' and ask for clarification.": The method column in the tables merely states "PDM", DPDM" etc. Please add cross-references (probably to a section in the appendix, and to a shorter summary in the main paper) to a detailed specification of exactly what the acronyms refer to (including citation, used variant of the general approach, type of noise augmentation, hyperparameters etc.) to make the experiments reproducible.

---

> > > ### Author Response · Authors · 2026-04-03
> > >
> > > 1. We agree and will include a general recipe for applying our method when the manifold is represented implicitly as the level set of some surface (arbitrarily restructuring a given constraint such that the right-hand-side is equal to zero will yield such a situation). We provide a brief formula for this situation here and will add it to the paper. For $h(x) = 0$ defining the manifold as we discuss in the paper, a procedure for applying our technique is as follows:
> > >
> > > 1 - For each $x^{i}$ in the training set, compute the Jacobian $J_{h}(x^{i}) \in \mathbb{R}^{k \times d}$ (either analytically through differentiation of $h(x)$ or through numerical or automatic differentiation techniques for more complicated $h(x)$)
> > >
> > > 2 - Compute an orthonormal basis of $J_{h}(x^{i})^{\top}$ to yield an orthonormal basis for $N_{x^{i}}\mathcal{M}$ and
> > >
> > > 3 - Draw noise $\xi \sim \mathcal{N}(0, \sigma^{2} I_{k})$
> > >
> > > 4 - Use basis vectors of $N_{x^{i}}\mathcal{M}$ and map $\xi$ onto the normal space to obtain $\tilde{\xi}$
> > >
> > > 5 - Construct the perturbed sample $x^{i}_{\sigma} = x^{i} + \tilde{\xi}$
> > >
> > > 6 - Train on the collection of perturbed samples $\{ x^{i}_{\sigma} \}$
> > >
> > > We will adjust Algorithm 1 in the paper accordingly in order to make it much clearer how our approach can be implemented for general implicit surfaces representing constraint manifolds of interest. This can also certainly be generalized to a set of constraints, so long as the constraint that applies at a particular point is still (first-order) differentiable, so the above procedure holds without loss of generality to the case where we have a set of implicit equations defining the manifold that the reviewer mentions. We thank the reviewer for this helpful suggestion and will continue to clarify the narrative of the paper to make it clearer to practitioners how our approach can be applied to general constraints.
> > >
> > > 2. This is certainly interesting and highly relevant work, and we thank the reviewer for ensuring that it is mentioned in our paper! We will include it in our discussion of related work in the paper, and we note that while their inflation step is highly similar to ours, their deflation step is different and requires analytic density correction that imposes additional requirements on the manifold and distribution being approximated. In particular, it appears to require (almost-everywhere) invertibility of the inflation step, which can be quite restrictive for general manifolds. In contrast, our alternative of manifold geometric post-projection uses straightforward nearest-point projection and optimization-based techniques to return samples to the manifold and form an intrinsic density. Our approach removes the need for invertibility of the inflation step while still yielding useful theoretical bounds. We also provide a simpler and more user-friendly implementation, with broader experiments and generality beyond the normalizing flow case and reduced restrictive assumptions needed for implementation.
> > >
> > > 3. We will include precise definitions and clarify the experimental setup that we provide in the appendix! We note that we do state that all diffusion model techniques use the same base architecture and training parameters unless otherwise specified in order to ensure as fair of a comparison as possible amongst the diffusion-based approaches. DDPM refers to the standard Denoising Diffusion Probabilistic Models, PIDM refers to Physics-Informed Diffusion Models, and PDM refers to Projected Diffusion Models. As of now, all of these acronyms were already defined earlier in the paper when they are first mentioned (either in the introduction or in the discussion of related work). To improve clarity, we will certainly expand our discussion of the details associated with each of the variants we experiment with and include precise hyperparameters associated with the training of each variant. Details will be further elaborated upon in the appendix, and we will clarify the necessary aspects of the experimental setup in the main body of the paper to improve understandability of our results for readers.
> > >
> > > We thank the reviewer for their detailed discussion and understanding of our paper! We will incorporate the useful suggestions made by the reviewer into the paper to improve clarity, presentation, and practical utility of our proposed technique. We hope that we have sufficiently addressed the reviewer's concerns and thank them deeply for their input, feedback, and guidance.

---

### Official Review · Reviewer_ykR4 · 2026-03-11

**Soundness:** 4
**Presentation:** 4
**Significance:** 3
**Originality:** 2
**Overall Recommendation:** 5
**Confidence:** 4

**Summary:**

This paper proposes a general preprocessing/postprocessing method called "Manifold-Aware Perturbations" to improve the ability of generative models to learn from constrained data. The core idea is to add Gaussian noise along the normal direction to the original manifold distribution p0, obtaining a non-degenerate perturbed distribution pσ. An arbitrary standard generative model is then used to learn pσ, and finally, the sampled results are projected back onto the manifold. Theoretically, the authors prove the non-degeneracy of pσ, exact recovery on linear manifolds, and an upper bound on the projection error for general manifolds in terms of the noise scale σ and the manifold's reach. Experiments cover multiple tasks ranging from planes, spheres, and mesh surfaces to MNIST images (with total flux constraint) and protein backbone generation. Comparisons with various baselines (DDPM, Glow, RealNVP, PDM, etc.) show that the method outperforms or matches existing approaches in distribution quality (COV, JSD, MMD) and sampling efficiency, while significantly reducing numerical instability during training (e.g., score explosion).

**Compliance With Llm Reviewing Policy:**

Affirmed.

**Final Justification:**

Weighing across dimensions:

Soundness (4): The theoretical derivations are rigorous, and the experiments are extensive across multiple tasks and models. The method is technically reliable.

Presentation (4): The paper is very clearly written and well-structured, making the motivation and algorithm easy to follow.

Significance (3): The problem of learning data on constrained manifolds is important for scientific applications. The method is simple, effective, and works with any generative model.

Originality (2): The components are existing tools, but their systematic combination to solve score explosion and distribution distortion is novel. Originality is fair.

Impact of the rebuttal: The authors responded honestly, clarifying the practical guidance for choosing the noise scale and acknowledging the limitation of requiring a known manifold. The rebuttal did not change my assessment, which remains an Accept.

Conclusion: The paper is technically solid, clearly presented, and addresses a meaningful problem. I recommend Accept.

**Key Questions For Authors:**

Questions:

1.	Experiments show that the choice of σ is critical to performance, and the optimal σ is related to the manifold's reach. For manifolds where the reach is unknown, are there heuristic rules or adaptive algorithms (e.g., based on data density estimation) to guide the selection of σ and avoid expensive grid searches?

2.	The method assumes the manifold has a known analytical form or a discrete mesh representation. For real-world data consisting only of point clouds without a reliable mesh, how can normals be estimated and projection implemented in practice? What impact does the deviation between normals estimated by local PCA and the true normals have on the final generation quality?

**Limitations:**

yes

**Strengths And Weaknesses:**

Strengths:

Data lying on low-dimensional manifolds is a common phenomenon in science and engineering, yet standard generative models struggle to handle such degenerate distributions directly. The proposed normal-noise-plus-projection framework is simple and effective in addressing this issue. Theoretically, the paper provides an explicit upper bound on the total variation distance between the projected and original distributions (Theorem 4.6), and proves exact recovery on linear manifolds and convergence as σ→0 on general manifolds. The theory is solid. Experimentally, the paper progresses from the simplest linear manifolds (planes) to highly nonlinear real-world problems; the baselines are diverse, including different generative models and their variants; the evaluation metrics cover distribution similarity, diversity, and computational efficiency, with clear and compelling results.

Weaknesses:

The method is essentially a pipeline of "add noise → train → project." Although clever, all components are existing tools, and the work lacks more disruptive theoretical or algorithmic innovation.

---

> ### Author Rebuttal · Authors · 2026-03-31
>
> We thank the reviewers for their highly insightful questions!
>
> 1 - We strictly consider the case where the constraint function h(x) with zero level set implicitly inducing the manifold surface is known analytically. As discussed later, this is frequently the case in a wide variety of scientific settings. As h(x) is assumed to be known in closed-form, it may be differentiated by a user analytically. The reach can be written as a bound with respect to 1st- and 2nd-order derivatives (see Aamari et al., “Estimating the Reach of a Manifold”, 2019 for more details, which is cited and recommended in our paper on line 181). Thus, a practitioner may use straightforward vector calculus to derive worst-case lower bounds for the reach, from which $\sigma$ can be estimated and our bounds may be applied.
>
> With regards to data density estimation, one may certainly do some amount of exploratory investigation in advance to determine whether the majority of the data density (a) lies on a low- or high-density region of M and (b) what the local reach at the regions of high data density is, through sampling points and/or local gradient methods. We remark that even for very small $\sigma$, our results generally were either competitive with the raw baseline or improved. Therefore, in general, while too-large $\sigma$ led to worsening results on manifolds of non-zero curvature, too-small $\sigma$ (as long as $\sigma$ was non-zero) rarely, if ever, harmed results. In conclusion, under a limited budget of computational resources (and assuming appropriate data normalization otherwise), one can select a very small $\sigma$ and gradually increase if improved results are needed.
>
> 2 - We do assume M has a known analytical form as our method is intended to combat the case where data is known to adhere to equality constraints (e.g. laws of physics). This is a major thrust of research in scientific machine learning, wherein domain experts wish to better incorporate known constraints into otherwise unknown samples obtained from generative models.
>
> We do note, however, that the other case where M is not given a priori certainly does exist in other applications. In follow-on work, we currently are investigating strategies for extending our method for the case where the normal vectors (and subsequent projection) can only be extracted approximately through other means and are investigating the downstream errors induced by such approximate noising and projection. However, our paper only considers the case where the constraint is known.
>
> For ease of understanding, we draw the connection in scope to PINNs, wherein the standard use case is for when the physics are known a priori (otherwise the inclusion of a physics-informed loss would be impossible).

---

> > ### Author Rebuttal · Reviewer_ykR4 · 2026-04-01
> >
> > I thank the authors for their thoughtful responses. My questions were largely exploratory, and the rebuttal clarifies the intended scope of the work. I do not see a need to change my original assessment and will maintain my score.

---

### Official Review · Reviewer_fhtM · 2026-03-12

**Soundness:** 2
**Presentation:** 3
**Significance:** 3
**Originality:** 3
**Overall Recommendation:** 4
**Confidence:** 4

**Summary:**

The authors study generative modeling when the data are forced to satisfy exact equality constraints, so the true data do not fill the whole ambient space, but instead lie on a lower-dimensional manifold. The authors do not propose a new diffusion architecture or a new flow architecture, but rather: (1) a distributional pre-processing trick (anisotropic noising), (2) a distributional post-processing trick (post-hoc manifold projection) and (3) a theoretical analysis giving exact recovery for linear constraints and a TV bound for nonlinear manifolds. The authors conclude with extensive numerical experiments ranging from simplistic to more realistic, and compare to existing approaches.

**Compliance With Llm Reviewing Policy:**

Affirmed.

**Final Justification:**

I appreciate the explanations of the authors. I'm willing to raise my score if they include a discussion of the theoretical limitations of their paper.

**Key Questions For Authors:**

1) How should the covariance matrix in the noising step be chosen?

2) How robust is the method when the manifold is only approximately known or when Jacobians and projections are numerically imperfect?

3) Can this framework be extended beyond exact equality constraints (approximate constraints, noisy constraints, inequality constraints, boundaries, corners, unions of manifolds, ...)?

**Limitations:**

The authors do not explicitly discuss limitations of their approach. Some limitations are

1) The method assumes that the constraint manifold are known well enough so that local Jacobians, normal directions and projections can be computed.

2) The theory assumes a fairly regular manifold: compact, C^2, closed with positive reach and with Lipschitz density on the manifold.

3) The choice of $\sigma$ matters: the sphere experiment confirms that too large of a choice of $\sigma$ relative to the reach affects performance.

4) The approach is restricted to the case where Lebesgue measure is the reference measure.

**Strengths And Weaknesses:**

Soundness:
The method is overall sound; the pre and post-processing allow the usage of standard diffusion and flow methods. On the other hand, Theorem 4.6 only bounds the discrepancy between the ideal projected law and the original law. In practice, however, one trains a diffusion or flow model only approximately, so the projected law is not the ideal one. As such, the bound in Theorem 4.6 does not capture the full end-to-end approximation error and may therefore be too optimistic as a practical guarantee.

Presentation:
The method is well explained overall, and the extensive numerical section is explained in detail. This is in part due to the fact that noising, projection as well as diffusion and flow models are already well known. On the other hand, the paper would benefit from a clearer explanation how the covariance matrix in the noising step is found.

Significance:
Both pre and post-processing steps are well chosen for equality-constrained distributions. However, the paper still relies on standard diffusion or flow methods and adds noising and projection as external steps. I therefore remain unconvinced that the method constitutes a major step forward over simpler projection-based baselines, even though the empirical results are often competitive and in some cases strong.

Originality:
The originality is limited. Projection and Gaussian noising are well known; the main novelty lies in choosing the perturbation in a manifold-aware way through the normal directions and in analyzing the resulting recovery error. This is a meaningful contribution but not a new generative modeling paradigm.

---

> ### Author Rebuttal · Authors · 2026-03-31
>
> We thank the reviewer!
> Many points focus on whether M being known is restrictive. Our method addresses the case where scientists wish to align generative models with domain knowledge.
> Regarding significance, we do not claim that we introduce a novel generative model, but rather clarify that we overcome known training and sampling issues in generative modeling in a principled, easy-to-implement way and provide corresponding theory. Our premise is that SOTA generative models struggle with constraints due to inherent mathematical limitations. We present a concise strategy which only introduces one hyperparameter, $\sigma$ (which can be roughly estimated using mathematical strategies) which allows these SOTA models to be used out-of-the-box with often dramatically improved performance.
> We agree with the reviewer that our method relies on standard diffusion or flow methods but emphasize that this is an intentional feature of our approach. Our goal is that scientists can (a) apply existing generative models while knowing that the generative pipeline is domain-informed and (b) not need to implement novel generative models. We hope that this clarification shows that the significance of our method is in its flexibility for drastically boosting the effectiveness of SOTA generative models in science, not as a novel generative model in itself.
>
> Responses to key questions:
>
> 1 - The covariance matrix may be written analytically. Let the orthonormal basis for $N_{z}M$ be $U_{N}(z)$. We can equivalently write $n \sim N ( 0, \sigma^{2} I_{N_{z}M})$ in the paper as
> \[N | z \sim N(0, \sigma^{2}U_{N}(z)U_{N}(z)^{\top}).\]
> $U_{N}$ can be computed directly from the Jacobian of $h(z)$, as the row space of this Jacobian is the normal space at $z$.
> In practice, we often find it more efficient to use other techniques (e.g. drawing from a random Gaussian and projecting it onto the normal space) than to form the covariance matrix, as these induce the same distribution.
> 2 - As discussed, the scope of this paper is where M represents constraints. However, the case where M is only approximately known is certainly possible. We note that knowledge of M itself is less important than knowledge of the normal directions to M, which is the true only need for implementation of the 1st stage of our method. Thus, slight offsets are okay as long as we roughly know local curvature, at least to 1st-order. In this setting, if practitioners are okay without postprojection and $\sigma$ is small, one could omit postprojection. If projections are numerically imperfect, our method should still be robust for small $\sigma$. If projections are not trustworthy, practitioners may have less need to realign samples with constraints, so for small $\sigma$, one can omit projection entirely.
> 3 - We discuss approximate and noisy constraints in 2. For inequalities, this would no longer represent a surface (thus manifold) in R^d. In general, boundaries/corners occupy a measure zero set with respect to both the m-dimensional Hausdorff measure as well as the d-dimensional Lebesgue measure, so the chances of a sample lying there are unlikely. In this case, a small perturbation along M should suffice. We do consider unions of M in the protein task, again showing that while we require mild regularity for theory, our method is highly flexible and truly only requires computing local (numerical or analytic) Jacobians. If manifolds are not disjoint and have intersections, the remarks about boundaries/corners apply.
>
> We have written a Limitations section discussing the known M requirement, $\sigma$ selection, computational cost, and the gap between theory (no training error) and practice. We address the reviewer’s suggested limitations here.
> 1 - We emphasize that our use case is for known M, allowing local Jacobians, normal directions, and projections to be computed either through optimization or in closed-form.
> 2 - We note that while these may appear restrictive, leading constrained/manifold generative modeling works are even more restrictive. For example, one could argue that the leading family of constrained generative models are Riemannian approaches:
> - de Bortoli et al., “Riemannian Score-Based Generative Models,” 2022
> - Mathieu et al., “Riemannian Continuous Normalizing Flows,” 2020
> - others
> These assume Riemannian manifolds (i.e. infinitely diff'able M). Our method requires (a) 2nd-order diff'ability for theory and (b) 1st-order diff'ability where $p_{0}(x) \neq 0$ for implementation. This is a dramatic relaxation and circumvents a great deal of computational necessities through extrinsic techniques. Thus, while we assume some regularity for theory, M can be complex or loosely defined and our method can still be implemented with minimal to no challenges.
> 3 - Addressed.
> 4 - The Lebesgue measure as the reference measure is standard for generative modeling and elementary calculus more broadly, as Lebesgue measure theory underpins differentiation and integration in R^d.

---

> > ### Author Rebuttal · Reviewer_fhtM · 2026-04-03
> >
> > Thank you for your answers and the clarifications regarding the positioning of the paper and covariance noising step.
> >
> > However, my main concern remains unresolved. The theoretical results consider discrepancies between the target and ideal projected law, whereas in practice one only has an approximately trained model and thus only an approximation of the target distribution before projection. While the rebuttal acknowledges this gap, as far as I can see, it does not provide a clear decomposition or stability argument explaining how pre-projection training errors propagate through the projection step.
> >
> > Regarding robustness and scope, the rebuttal provides plausible intuitions. I appreciate this discussion, but at present it still reads mostly as heuristic guidance rather than as a developed theoretical or empirical treatment within the paper.

---

> > > ### Author Response · Authors · 2026-04-03
> > >
> > > We thank the reviewer for their in-depth understanding of the paper!
> > >
> > > The issue of training error is certainly relevant, but this is an issue in any work where diffusion models or normalizing flows are employed. In fact, we would argue or claim that due to the numerical instabilities arising from the ground truth score field being (numerically) infinite at small t near the manifold, the $p_{\sigma}(x)$ distribution for nonzero $\sigma$ is easier to approximate than the raw $p_{0}(x)$ when $p_{0}(x)$ is indeed constrained. Therefore, training error should be reduced when (for example) a diffusion model is tasked with learning $p_{\sigma}(x)$ instead of $p_{0}(x).
> > >
> > > Training and approximation error in generative models more broadly is certainly an interesting direction of work, but at present, we consider it to be a separate line of research that perhaps those in ML or approximation theory might be better suited to study. For theoretical discussion, we confine the scope of our paper and assume that the generative model being employed is indeed a distribution approximator. Our paper simply uses a manifold-aware perturbation to strategically changes the distribution that is being approximated by an arbitrary generative model so that it is easier to approximate and later sample from. Theoretical treatment of training errors, while certainly important, are less within the scope of our work. However, we absolutely agree with the reviewer that this is important to discuss within the paper as an important potential limitation of the theoretical results we present and we will address it within the main body of the text.
> > >
> > > Based on the reviewer's recommendations, we will use the intuition and elucidation written in our rebuttal and incorporate it into the main body of the paper. We hope that this will help reframe the current heuristic guidance into the precise theoretical relationships between the reach and manifold curvature and $\sigma$ that we intended to convey in our writing of the paper. We hope that this helps resolve the reviewer's concerns regarding robustness and scope. Thank you very much!

---

### Decision · Program_Chairs · 2026-04-30

**Decision:**

Accept (spotlight)

**Comment:**

This paper proposes a method to train generative models when the data distribution is supported on a set given by equality constraints. The authors first add noise only on the normal space of the manifold, train the model, and finally project samples onto the manifold. Although some reviewers saw the fact that the equality constraints need to be known beforehand as a weakness, they all praised the theory and the experiments in the paper, and all had a positive assessment. This paper is a clear accept.